# Advances in Chemical Composition, Extraction Techniques, Analytical Methods, and Biological Activity of *Astragali Radix*

**DOI:** 10.3390/molecules27031058

**Published:** 2022-02-04

**Authors:** Xiangna Chang, Xuefeng Chen, Yuxi Guo, Pin Gong, Shuya Pei, Danni Wang, Peipei Wang, Mengrao Wang, Fuxin Chen

**Affiliations:** 1College of Food and Biological Engineering, Shaanxi University of Science and Technology, Xi’an 710021, China; changxiangna@sust.edu.cn (X.C.); chenxf201693@163.com (X.C.); guoyuxi416@163.com (Y.G.); p14753269@163.com (S.P.); wangdanni@sust.edu.cn (D.W.); wangpp816@163.com (P.W.); 13488311504@163.com (M.W.); 2School of Chemistry and Chemical Engineering, Xi’an University of Science and Technology, Xi’an 710054, China; chenfuxin1981@163.com

**Keywords:** *Astragali Radix*, extraction technologies, chemical constituents, analytical methods, biological activity

## Abstract

*Astragali Radix* (AR) is one of the well-known traditional Chinese medicines with a long history of medical use and a wide range of clinical applications. AR contains a variety of chemical constituents which can be classified into the following categories: polysaccharides, saponins, flavonoids, amino acids, and trace elements. There are several techniques to extract these constituents, of which microwave-assisted, enzymatic, aqueous, ultrasonic and reflux extraction are the most used. Several methods such as spectroscopy, capillary electrophoresis and various chromatographic methods have been developed to identify and analyze AR. Meanwhile, this paper also summarizes the biological activities of AR, such as anti-inflammatory, antioxidant, antitumor and antiviral activities. It is expected to provide theoretical support for the better development and utilization of AR.

## 1. Introduction

*Astragali Radix* (AR), known by the Chinese name of Huang-qi (Ougi in Japanese name), is a traditional herbal medicine that has been used for over 2000 years in China [1]. According to the Pharmacopoeia of the People’s Republic of China (2020 edition), AR is defined as the dried root of *Astragalus membranaceus* (Fisch.) Bge. or *Astragalus membranaceus* (Fisch.) Bge. *var. mongholicus* (Bge.) Hsiao (*Fabaceae*) [1], which is a variant of *Astragalus membranaceus* (Fisch.) Bge. and mainly grows in Shanxi, Inner Mongolia, Shaanxi, Gansu and other provinces in China as well as Korea and Mongolia [2]. It was found that the main bioactive compounds in AR samples from different suppliers varied greatly [3]. Therefore, a standard is needed to evaluate the quality of AR. Sun et al. carried out a principal component analysis of 18 kinds of AR comprehensive indicators that can be combined with diameter, alcohol (water) extract, flavonoid aglycone and aglycone peak area ratio to establish an AR grade evaluation system to provide analysis methods [4]. In modern pharmacological research, AR has been demonstrated to possess tonic, hepatoprotective, diuretic, and expectorant properties and exhibit antihyperglycemic, immunomodulating, anti-inflammatory, antiviral and antioxidant activities [5,6,7], and so on. Nowadays, AR is used extensively in clinical therapy, such as, Huangqi injections for treating renal diseases (Figure 1). However, the chemical compounds and active constituents of AR remain obscure.

Over the years, extensive research has been conducted on the chemical components of AR. It is known that AR root contains saponins, polysaccharides, flavonoids, amino acids, and trace elements [8]. Among these, saponins are the major active constituents, especially Astragaloside IV (AS-IV). However, the composition and quantity of active compounds may vary depending on the culture area, the growth period and the growth conditions. In addition, the compounds identified from crude drugs also depend on the method of extraction and the method of analysis. Nevertheless, to date, there are few reviews on the extraction methods and analytical methods for AR chemical components. Thus, this review summarizes the recent advances in the chemical studies of AR species, involving different extraction techniques and qualitative and quantitative analytical methods, and provides a summary of their biological activities.

## 2. Chemical Composition

According to available studies, more than 100 compounds have been isolated and identified from AR. Based on structure, it can be mainly divided into four groups, including polysaccharides, saponins, flavonoids and others. Enhancing the classification of different compounds in AR might facilitate the understanding of the pharmacological effects of AR.

### 2.1. Polysaccharides

The polysaccharides from AR are well-known for their hepatoprotective effects [9]. As early as 1982, two glucans, AG-1 and AG-2, and two heteropolysaccharides, AH-1 and AH-2, were isolated from the aqueous extract of *A. membranaceus*. AG-1 is a water-soluble *α* (1→4) (1→6) glucan, and the composition ratio of *α* (1→4) (1→6) glycosidic glycosyl groups is 5:2. AG-2 is a water-insoluble *α* (1→4) glucan. AH-1 is a water-soluble acidic polysaccharide. After hydrolysis, hexuronic acid, glucose (Glc), rhamnose (Rha) and arabinose (Ara) uniting at 1:0.04:0.02:0.01 were detected. AH-2 was water soluble, with Glc and Ara uniting at 1:0.15, detected after hydrolysis. The polysaccharides APS-I and APS-II were isolated from the water extract of *A. membranaceus*. APS-I consisted of Ara and Glc, in a molar ratio of 1:3.45 and APS-II consisted of Rha, Ara and Glc uniting at 1:6.25:17.86 [7,10]. Fang et al. produced APS-III by water extraction and alcohol precipitation from *A. membranaceus* [11]. They were all water-soluble and insoluble in alcohols and other organic solvents.

The analysis of the polysaccharide from AR revealed three sugars: Glc, Rha, and Ara. Among them, Glc was the major component, followed by Ara and Rha [12]. Table 1 lists the parameters of the identified polysaccharides, such as molecular weights (*Mw*), identification methods, classifications, sources, and reference.

### 2.2. Saponins

A lot of saponins have previously been isolated and authenticated from AR. In 2007, Xu et al. isolated and identified Astragaloside I (**1**), Astragaloside II (**4**), Astragaloside IV (**6**), Astragaloside IV isomer/III (**7**) and Astragaloside V/VI/VII (**8**,**9**,**10**) in the roots of *A. membranaceus* using high-performance liquid chromatography–tandem atmospheric pressure chemical ionization mass spectrometry (HPLC–APCI–MS) [15]. In addition, the compounds Soyasaponin II (**11**) and Soyasaponin I (**12**) were detected and analyzed in AR using high-performance liquid chromatography-electrospray ionization-quadruple-time of flight-mass spectrometry (HPLC–ESI–QTOF/MS) [16]. Moreover, Agroastragaloside I (**15**) was found by Hirotani et al. [17]; Agroastragaloside III (**13**) and Agroastragaloside IV (**14**) were determined by Zhou’s group [18]; Kitagawa et al. authenticated Isoastragaloside I (**2**), Isoastragaloside II (**5**) and Acetylastragaloside I (**3**) with NMR [19]. These compounds all belonged to agroastragaloside derivatives. Except saponins compounds **1, 4, 6,** and **13**, Yu et al. also obtained alexandroside (**16**) from AR stem and leaves [20].

Figure 2 and Table 2 give a detailed information regarding 16 saponins identified from AR including their structure, molecular formulas, structural formulas, *Mw*, parent ions, fragment ions, sources, and others.

### 2.3. Flavonoids

Until now, many flavonoids have been isolated and identified from AR. Formononetin (**1**), ononin (**2**) and calycosin-7-*O*-β-d-glycoside (**6**) were separated from AR and analyzed using UPLC–DAD–ELSD [20]. In 2013, formononetin-7-*O*-β-d-glucoside-6”-*O*-acetate (**4**), calycosin-7-*O*-β-d-glucoside-6”-*O*-acetate (**8**), pratensein (**9**), pratensein-7-*O*-β-d-glycoside (**10**), biochanin-A (**11**) and (3*R*)-7,2′-dihydroxy-3′,4′-dimethoxyisoflavan (**12**) were isolated rapidly and identified with the help of HPLC–APCI–MS/MS [23]. Lin et al. (2000) isolated and obtained the constituents formononetin-7-*O*-β-d-glucoside-6”-*O*-malonate (**5**), calycosin-7-*O*-β-d-glucoside-6”-*O*-malonate (**7**), (6a,llaR)-3-hydroxy-9,10-dimethoxypterocarpan (**13**), (6a,llaR)-3-hydroxy-9,10-dimethoxypterocarpan-3-*O*-β-d-glycoside (**14**) and astraisoflavanglucoside-6”-*O*-malonate (**15**) using LC–ESI–MS [24]. Du et al. (2014) obtained and confirmed calycosin (**3**), wogonin (**16**), 3-hydroxyflavanone (**17**), medicarpin (**18**), isorhamnetin (**19**), mangiferin (**20**) and naringin (**21**) with the help of liquid chromatography–tandem mass spectrometry (LC–MS) from AR [16].

The structures of the flavonoids reported in this review are illustrated in Figure 3 and Table 3 with their corresponding information, including molecular formula, structural formula, molecular weight, parent ion, fragment ion, source, etc.

### 2.4. Others

Except the active compounds referred to above, AR also contains numerous other constituents, such as amino acids, trace elements (Fe, Cr, Zn, Se, Mn, Ag, Co, Cu, and so on), active small molecules—riboflavin, caffeic acid, chlorogenic acid, niacin, linoleic acid, coumarin, folic acid, linolenic acid, lupeol, daucosterol, vitamin P, starch E, sterols, betaine, and choline—and other components. The chemical structures and molecular formulas of parts of these compounds are shown in Figure 4.

So far, it has been demonstrated that there are more than twenty kinds of biogenic amino acids in AR, such as glycine, *γ*-aminobutyric acid, asparagine, cystine, aspartic acid, threonine, glutamic acid, valine, alanine, methionine, isoleucine, serine, leucine, and so on. Among them, lysine, threonine, valine, leucine, methionine, isoleucine, and phenylalanine are essential amino acids for the human body [25].

## 3. Extraction Methods for AR Components

Since the components in AR can be divided into macromolecule compounds and small molecules, we discuss the different extraction methods used in the exploration of AR components mainly from two aspects: macromolecules (polysaccharides) and small molecules (saponins and flavonoids).

### 3.1. Extraction Methods for Astragalus Polysaccharides (AP)

*Astragalus* polysaccharides are a kind of bioactive ingredient extracted from *Astragalus membranaceus*, mainly composed of fructose, rhamnose, arabinose, hexuronic acid, glucose, galacturonic acid and glucuronic acid [25]. The difficulty in obtaining polysaccharides is breaking the cell wall and promoting the release of polysaccharides from the cytosols. To date, the methods used for extracting AP mainly consist of water extraction, microwave-assisted extraction, ultrasonic extraction, and enzyme extraction [26].

#### 3.1.1. Water Extraction

*Astragalus* polysaccharides are soluble in water and insoluble in alcohol [27]. At present, the extraction of AP mainly adopts water extraction technology, which is simple and easy to operate. However, the energy consumption is large during the process, and the purity of extracted polysaccharides is not high.

Applying statistics-based experimental designs to optimize the extraction process of AP. The optimal conditions were: extraction time of 3 h, liquid-to-solid ratio of 4, and particle size of 33.8 mesh. Under these conditions, the maximal extraction rate of AP was 16.32% [27]. The water reflux extraction technology of AP was optimized, the ratio of solid to liquid, extraction time, extraction times and immersion duration were selected to be experimental factors, and the content of AP was used as the inspection index. The optimum extraction process conditions were: ratio of solid to liquid—1:12; extraction time—1 h; immersion duration—1 h; and number of extractions—3. Under these conditions, AP accounted for 6.4% of the original herbs [28].

#### 3.1.2. Microwave-Assisted Extraction

The microwave has a good auxiliary effect on the extraction of AP. The thermal effect of microwaves can break the cell wall [29]. Cellular polysaccharides can be easily extracted after breaking through the cell wall and cell membrane barriers.

The microwave-assisted extraction method for AP was studied, with water as the extraction solvent. The optimal extraction conditions were: the ratio of water/material— 12:1; pH—9.0, regulated by saturated limewater; microwave power—300 W; 2 extractions, each 10 min. The final yield was 14.6% and the purity was 88.1% [30]. Taking a yield of AP as the inspection index and water as the extractant, an orthogonal design experiment was used to study the microwave extraction process of AP. The best extraction conditions were: 30 times for water and 3 times for microwave extraction—, each 10 min [31].

Microwave extraction technology was also used to extract AP from the dried roots of *Astragalus membranaceus*. The extraction time, heating temperature, pH value of extraction solution and solid–liquid ratio were optimized. The optimal operating process parameters were determined as follows: microwave heating time—20 min; heating temperature—120 °C; solid–liquid ratio—8:1; and pH value—10. The crude polysaccharide yield was 32% and the purity was 44.4% [32].

#### 3.1.3. Ultrasonic Extraction

Ultrasonic extraction utilizes ultrasonic waves to convert electromagnetic energy into thermal energy, makes openings and cracks appear on cells, and then breaks the plant cell wall and cell membrane structure, finally resulting in the release and dissolution of intracellular substances [33]. Compared with traditional extraction methods, such as decoction, ultrasonic extraction has the advantages of shortening the extraction time, increasing the extraction efficiency, and increasing the leaching rate of the target component.

Using the ultrasonic extraction method to extract polysaccharides, the optimum extraction conditions of AP in the single-factor test were: λmax = 490 nm, particle fineness—0.25 mm~0.5 mm; ultrasonic time—25 min; ratio of solid to liquid—1:25(vol); and extraction temperature—30 °C. The content of AP extracted could reach 4.92%.

#### 3.1.4. Enzyme Extraction

Cellulose, the main component of AR cell walls, is the main barrier used to prevent the release of macromolecules such as intracellular polysaccharides. The traditional enzyme used to extract *Astragalus* polysaccharides is cellulase [34], which can hydrolyze the cell wall and dissolute intracellular components to increase the extraction rate [35].

The combined enzyme extraction of *Astragalus* polysaccharide was optimized. A response surface methodology and orthogonal tests were applied to optimize the conditions: the combined enzyme concentrations of cellulase, pectinase, and papain were 1.5%, 1%, and 0.5%, respectively; the extraction time was 94.5 min; the extraction temperature was 49.9 °C; the pH value was 5.1 [26].

A response surface method was used to optimize the glucose oxidase-assisted extraction process to obtain the maximum yield of crude APS. The optimized extraction conditions were as follows: enzyme dosage, 3.0%; enzyme treatment time, 3.44 d; enzyme treatment temperature, 56.9 °C; extraction solvent pH of 7.8. Under these conditions, the experimental yield was 29.96 ± 0.14% [36]. The conditions of the enzyme extraction method are mild, the process is simple and yield is high; nevertheless, the enzyme is expensive.

### 3.2. Extraction of Saponins and Flavonoids

Saponins and flavonoids are the major active constituents in AR. Saponins mainly consist of astragaloside IV, astragaloside I and soyasaponin I and others; flavonoids mainly include calycosin, formononetin and ononin and others. The study of the extraction methods of saponins and flavonoids in AR has a positive effect on the screening of high-quality *Astragalus* resources and the identification of AR medicine quality. Presently, the main extraction methods of saponins and flavonoids from AR include reflux extraction, microwave-assisted extraction, and enzymolysis-assisted extraction [24].

#### 3.2.1. Ethanol Reflux Extraction

Ethanol reflux extraction is applied as the main method for extracting saponins and flavonoids currently. This method is simple, stable, and feasible, and more importantly, it can extract active constituents such as saponins as much as is possible in AR.

Yang et al. (2011) optimized the ethanol reflux extraction procedure parameters (ethanol concentration, the amount of ethanol added, extraction time, and extraction times) using an orthogonal design method, with the astragaloside IV extraction rate as the detection index. The optimal conditions of alcohol extraction were: 60% ethanol; ratio of solid/liquid—1:6 (g/mL); and 3 extraction cycles, each 1 h [37]. High-performance liquid chromatography (HPLC) was used to estimate the yield of flavonoids extracted from *Astragalus mongholicus* by the alcohol reflux extraction method. According to the results of single-factor experiments, the optimum conditions were confirmed as: temperature—75 °C; time—2.5 h; ethanol concentration—90% (*v*/*v*); and ratio of solvent to raw material—20 mL/g. The best yield was 0.934 mg/g [38]. Orthogonal tests were used to optimize the extraction conditions of total flavonoids from *Astragalus.* The contents of total extracts and flavonoids were regarded as indicators. The optimal process for the ethanol reflux extraction of total flavonoids from *Astragalus* was: extracted with 10 times 70% ethanol two times for 60 min each time [25].

#### 3.2.2. Microwave-Assisted Extraction (MAE)

An MAE method was used to extract astragalosides I-IV from AR. MAE gave the highest extraction efficiency within the shortest time. The optimal conditions of MAE were: employing 80% ethanol as solvent; ratio of ethanol to the material—1:25 (g/mL); temperature—70 °C; irradiation power—700 W; and 3 extractions, each 5 min [39]. The flavonoids were extracted with a microwave digestion system and the extraction rate of flavonoids was determined by a spectrophotometer. The optimum extraction conditions of flavonoids from *A**. membranaceus* according to the extraction rate of flavonoids were obtained as followed: concentration of ethanol—95%; extraction time—20 min; ratio of material to liquid—1:15 g/mL; and temperature—90 °C. Under these conditions, the extraction rate of flavonoids was the highest (0.489%) [26].

#### 3.2.3. Enzymolysis-Assisted Extraction

The active substances of plant-based herbs are often encased in lignins, and enzymes can effectively degrade lignin and greatly increase the extraction rate of traditional Chinese medicine.

An orthogonal design was used to study the optimum technology for the enzyme-assisted extraction of total flavonoids from *Astragalus.* The optimum extraction process was determined as follows: enzymolysis time—120 min; hydrolysis temperature—30 °C; enzyme dosage—8 mg; and pH—4.5 [40].

A response surface methodology was used to optimize the process of cellulase-assisted extraction of total saponins from *Astragalus*, and the optimal conditions were: pH—4.38; enzymolysis time—132 min; enzymolysis temperature—51 °C; enzyme dosage—8.84 mg; predictive extraction rate—1.89%; and actual extraction rate—1.87 ± 0.03%. The technological conditions for the extraction of total flavonoids from *A. membranaceus mongholicus* by cellulose hydrolysis-assisted ethanol extraction were studied. The optimum process parameters were: amount of added cellulase—7.5 μL/g; enzymatic hydrolysis pH value—4.0; enzymolysis temperature—55 °C; enzymolysis time—1.5 h; ratio of liquid to solid—30 mL/g; ethanol concentration—90%; extraction temperature—75 °C; and extraction time—2.5 h; under these conditions, the extraction rate of total flavonoids was 0.57%. It increased by 46.15% compared to traditional alcohol extraction (0.39%) [41].

Based on the above review, we compared the advantages and disadvantages of each extraction method as shown in (Table 4).

### 3.3. Others

Pan et al. used a two-phase solvent system of n-hexane ethyl acetate–ethanol–water (3:5:3:5, *v*/*v*) to purify mullein with a purity of 95.8% and a recovery rate of 85.9% [42]. Using the dissociation constant of the drug, Wang et al. used a liquid–liquid equilibrium system to separate multiple active ingredients: isomucronulatol 7-*O*-glucoside, astraisoflavan-7-*O*-β-d-glucoside, and calycosin-7-glucoside [43].

## 4. Analytical Methods

A number of methods have been developed for the identification and qualitative and quantitative analyses of AR, including spectrometry, all kinds of chromatographic methods such as thin-layer chromatography, high-speed countercurrent chromatography, high-performance liquid chromatography and their coupling techniques [44].

### 4.1. Spectral Analysis

Various materials composed of different structures have their own characteristic spectra. Spectral analysis is based on their characteristic spectra and used to study the structures of substances or to determine the chemical compositions.

The content of polysaccharides in *A. membranaceus* can be determined by spectrophotometry. Glucose and phenol-concentrated sulfuric acid were used as a reference substance and color reagent, respectively. The absorbance of the sample solution was measured under λ = 486 nm during the process. Compared to *Astragaloside* IV, the total contents of astragalosides in *Astragalus* grown in different areas were determined by ultraviolet spectrophotometry. The results showed that the total contents of astragalosides from Shanxi *Astragalus* were the highest (3.307 mg/g) [43]. Inductively coupled plasma atomic emission spectrometry (ICP-AES) was used to determine the contents of seven trace elements, Ca, Mg, Fe, Cu, Mn, Cr and Zn, in 46 *Astragalus* samples from Shanxi, Gansu, Neimeng and Sichuan. The results showed that there was almost no difference in the contents of these seven metal elements in the *Astragalus* samples from these four regions. It is worth mentioning that surface-enhanced Raman scattering (SERS) technology was used to distinguish between the authentic *Astragali Radix* (44 *Astragalus membranaceus* and 39 *Astragalus mongholicus*) and the most common adulterant samples (41 *Astragalus lehmannianus*) with a high sensitivity (100%) and specificity (97.6%) [45].

### 4.2. Chromatographic Analysis

With the selective distribution of different substances in different phases, chromatographic analysis elutes with the mixture in the stationary phase relative to the flow. Different substances in the mixture move at different speeds along the stationary phase and eventually reach separation.

Eight flavonoids (9,10-dimethoxypterocarpan 3-*O*-β-d-glucoside, ononin, 2′,4′-dimethoxy-3′-hydroxyisoflavan 6-*O*-β-d-glucoside, 10-hydroxy-3,9-dimethoxypterocarpan, afromosin, calycosin, formononetin, and rutin) in the *A. membranaceus* were separated and determined by using micellar electrokinetic chromatography (MEKC) with a DAD. Using fused silica capillaries, the micellar phase composed of 25% (*v*/*v*) acetonitrile, 100 mM sodium cholate, 20 mM Na_2_B_4_O_7_ and 20 mM H_3_BO_3_ buffer (pH 9.2) was used to obtain the optimal resolution at +25 kV and 25 °C [46]. A thin-layer chromatography (TLC) method for the determination of astragalin in *AR* was established, using a silica gel G thin board. Chloroform–methyl alcohol–water (13:6:2) was used as the developer. The detected wavelength and reference wavelength were at 530 nm and 700 nm, respectively. The results indicated that the linear relationship of astragalin curves was good. The average recovery rate was 98.72% with RSD = 3.5% [22].

### 4.3. Coupling Techniques

Modern high-tech and various chemical statistics methods play an important role in quality research and quality control of traditional Chinese medicines. Chromatography and its coupling technologies are one of the most important technological platforms for this research as a modern high-performance separation analysis technique.

Astragaloside I and astragaloside II were identified by thin-layer chromatography–tandem mass spectrometry (TLC–MS/MS) in the ethyl acetate extract of *Astragalus membranaceus.* They were isolated by means of high-speed countercurrent chromatography (HSCCC) with a two-step two-phase solvent system of ethyl acetate-2-propanol-water (5:1:5, 50:1:50, *v*/*v*/*v*). The contents of astragaloside I and astragaloside II were detected as 30.2 mg/g and 16.5 mg/g crude extract, respectively [47].

A sensitive and rapid ultra-performance liquid chromatography with tandem mass spectrometry (UPLC–MS/MS) method was established for the simultaneous quantitation of two major categories of bioactive substances (triterpenes and flavonoids) in *Astragalus* [48]. Liu et al. established an UPLC–MS method for the simultaneous determination of 4 saponins and 8 flavonoids, which can be used to analyze the effects of organic acid directed processing on the contents of astragaloside and aglingin [49].

Liu et al. (2014) identified the structures and determined the contents of 16 compounds (12 flavonoids and 4 astragalosides) from AR using ultra-performance liquid chromatography–electrospray ionization–tandem mass spectrometry (UPLC–ESI–MS) and high-performance liquid chromatography–diode array detector–evaporative light scattering detector (HPLC–DAD–ELSD), respectively [50]. An UPLC–QQQ–MS method for the simultaneous determination of 17 representative components in Astragali Radix-Curcumae Rhizoma was developed, which provides a feasible method for the holistic quality control of preparations [51].

## 5. Biological Activities of AR

*Astragalus,* as a traditional Chinese medicinal material, has been used in clinical treatment. The active ingredients in *Astragalus* are mainly saponins, polysaccharides and flavonoids. In addition, there are some other extracts, such as chlorogenic acid, caffeic acid, β-sitosterol and so on. Studies have found that *Astragalus* has many biological activities, such as anti-inflammatory, antioxidant, antitumor, antiviral, cardiovascular disease prevention, and anti-diabetic activities and so on [7,52,53] (Figure 5).

### 5.1. Anti-Inflammatory Activity

LI et al.’s [54] study found that Astragaloside IV (AS-IV) treatment significantly reduced the production of inflammatory cytokines in orbital fibroblasts induced by IL-1β in vitro, thereby inhibiting autophagy and preventing Graves’ orbital disease. Zhu et al.’s [55] study found that AS-IV was used to act on human umbilical vein endothelial cells (HUVEC), and then act on ox-LDL, and found that the ROS and NADPH oxidase activities of the cells were significantly reduced compared with the control group. At the same time, the expression levels of Nrf2, HO-1, TNF-*α* and IL-6 were significantly reduced. This shows that astragal side IV can protect oxidase low-density lipoprotein (OX-LDL)-induced endothelial cell damage and inhibit atherosclerosis by reducing oxidative stress and inflammation. In addition, other ingredients in *Astragalus* are also believed to have anti-inflammatory effects. Ruby et al. [56] treated lipopolysaccharide-induced obese mice with active ingredients (Rx) extracted from *Astragalus* and found that the activation of NF-κB in macrophages was dose dependent with Rx. The mRNA expression levels of the inflammatory cell markers CD68 and F4/80, as well as the expression levels of the cytokines MCP-1, TNF-*α* and IL-6, decreased significantly, which shows that *Astragalus* can reduce the symptoms of glucose intolerance, insulin resistance and hypertriglyceridemia caused by obesity by anti-inflammatory. Studies have found that these active ingredients include calyx-7-bD-glucoside (0.9%), onionin (1.2%), calyxin (4.53%) and formononetin (1.1%). Xuehong Nöst et al. [57] combined in vitro cytokine production analysis and LC–MS metabolomics technology to prepare extracts of different polarities from Huangqi Jianzhong Decoction. It was found that these ingredients can reduce the expression levels of TNF-*α*, IL-1β and IFN-*γ* in U937 cells, proving the anti-inflammatory effect of *Astragalus*; active ingredients such as verbascoside, verbascoside, astragaloside IV, glycyrrhizin, 18 β-glycyrrhetinic acid, paeoniflorin and leucoflorin were identified. The anti-inflammatory mechanisms of AR are presented in Figure 6.

### 5.2. Antioxidant Activity

Antioxidants have many positive effects on human health, protecting the body and cells from ROS and free radical damage and preventing the oxidation of biological macromolecules such as DNA, membrane lipids and proteins [7]. As a result, over the last few years there has been an increasing interest in developing safer and more effective natural antioxidants to prevent oxidative damage in the food and pharmaceutical industries [7].

The antioxidant activity of AR has been extensively studied. The antioxidant activity mechanisms of AR are presented in Figure 7. Ming et al. [58] used astragaloside IV to act on H_2_O_2_-induced retinal ganglion cells and found that the level of ROS in the cells decreased, and the mitochondrial membrane potential decreased, resulting in cells from oxidative stress damage, indicating that astragaloside IV has an antioxidant effect. Yang et al. [59] found that astragaloside IV has a protective effect on ischemia-reperfusion injury, Through the antioxidant and anti-inflammatory effects of *Astragalus*, it regulates the expression of Nrf2 and Bach1 protein in the nucleus, promotes the expression of HO-1 protein, and provides a protective effect. At the same time, Isamu Murata et al. [60] studies found that astragaloside IV can prevent muscle cell damage caused by crush syndrome through indirect antioxidant effects. Among them, astragaloside IV acts as a NO donor in injured muscles and improves mitochondrial dysfunction and inflammation.

Du et al. [22] established an streptozocin (STZ) rat model and then compared ginsenoside rg1 (group G), astragaloside IV (group A), ginsenoside rg1 and astragaloside IV combined (group C) administration groups, and observed the level of oxidative stress in rats. It was found that the oxidative stress levels of the three groups were significantly reduced, of which the total antioxidant capacity (T-AOC) of group C was the highest. Hyun-A et al. [61] studied the antistress effects of *Astragalus* through a fixed stress model in mice and found that the antianxiety effects of *Astragalus* extract (500 mg/kg), astragaloside IV (20 mg/kg) and buspirone (1 mg/kg) are comparable. Verbasin has been verified to have antioxidant activity. Verbasin can inhibit the production of ROS by enhancing the activity of antioxidant enzymes, namely glutathione peroxidase, catalase and superoxide dismutase (SOD) [62].

### 5.3. Anti-Tumor Activity

With the increasing side effects of chemotherapy on the treatment of tumors, the use of Chinese herbal medicine to treat tumors has become a new direction. Many studies have shown that *Astragalus* has antitumor effects [63]. The antitumor activity mechanisms of AR are presented in Figure 8.

Li et al. [64] found that APS can activate macrophages, releasing NO and TNF-*α*, thereby directly preventing the growth of cancer cells, and has antitumor effects on breast cancer mediated by macrophage activation. Cho et al. [65] isolated five biologically active components from the roots of *A. membranaceus*, and found that the components designated as AI are the most effective for the mitogenic effect of mouse spleen cells. Among them, AI can induce the monocyte differentiation of human and murine cells in vitro. AR administered in vivo can even partially restore the suppressed mitotic response in tumor-bearing mice. In addition, there are many studies on the antitumor effects of *Astragalus* and other Chinese herbal medicines. Li et al. [66] believe that polysaccharide peptides (PSP) and AP can be used as a new compound (PSP + APS) to assist adriamycin (AMD) chemotherapy. Through the establishment of a solid tumor model in mice, it was found that PSP+APS can significantly increase the percentage of CD3(+) and CD4(+) T lymphocytes in tumor tissues, the ratio of CD4(+)/CD8(+), and IL-2 in spleen and Bax/IL-2R expression. It can reduce the expression of Bcl-2 and CDK4 in tumor tissues.

Xu et al.’s [67] study found that different ratios of *Astragalus*-Edocarium extracts have inhibitory effects on the growth of Lewis lung cancer (LLC) in transplanted mice. Compared with the control group, *Astragalus*-Curcuma extract can significantly reduce tumor weight and tumor microvessel density (MVD), and the 3:1 treatment group has a similar MVD reduction effect to the cisplatin group. The study also found that the expressions of p38 MAPK, P-p38 MAPK, ERK1/2, P-ERK1/2, JNK, and P-JNK in the *Astragalus*-Edocarium treatment group were reduced compared to the control group. The expression of VEGF protein in the 2:1 and 3:1 treatment groups was significantly lower than that of the control group.

### 5.4. Antiviral Activity

*Astragalus* total saponins have been shown to have antiviral effects. Wang et al. [68] used astragaloside IV to act on the human HBV-transfected liver cell line HepG2 and found that hepatitis B surface antigen and e antigen were inhibited. This shows that astragaloside IV has an effective anti-HBV activity. Tang et al. [69] studied the treatment of chronic hepatitis B with the compound *Astragalus* and found that the clinical efficacy of the AC treatment group was significantly better than that of the control group. The negative seroconversion of hepatitis B virus (HBV) antigen e and HBV DNA was also significantly higher than that of the control group. At the same time, a duck hepatitis B model was established and verified, and the findings were consistent with the clinical results. This shows that AC can promote the recovery of viral hepatitis and inhibit HBV replication. Kajimura et al.’s [70] study found that the oral administration of *Astragalus* has a protective effect on Japanese encephtis virus (JEV) infection in mice. After applying *Astragalus* extract (AE), the number of mouse peritoneal exudative cells (PEC) (especially macrophages) increased significantly. The protective effect of oral AE is based on a non-specific mechanism. In the early stages of infection, before turning to antibody production, macrophages play an important role in resisting JEV infection, for example, by inducing the production of reactive oxygen species (AO). Wang et al. [71] found that, in the Hep-2 cell system, *Polygonum cuspidatum* and *Astragalus* have a synergistic (1:1) effect, which inactivates the HSV-1F strain and can inhibit the proliferation of HSV-1F and block the infection of HSV-1F. The antiviral effect of *Astragalus* was verified.

### 5.5. Cardiovascular Disease Prevention

An increasing number of experiments have confirmed the effectiveness of AR in inhibiting cardiovascular diseases such as myocardial ischemia-reperfusion injury [72], myocardial hypertrophy [73], vascular endothelial dysfunction [74,75], coronary artery disease [76], atherosclerosis [77], myocardial fibrosis [78], and viral myocarditis [79].

The sarcoplasmic reticulum (SR) plays an essential role in intracellular calcium regulation, and the sarcoplasmic reticulum Ca^2+^- ATPase (SERCA2a) in the SR is responsible for sarcoplasmic calcium reuptake. Many studies have shown that the expression of SERCA2a is significantly reduced after myocardial injury. AR has been reported to promote SERCA2a expression, enhance calcium reuptake in the SR and inhibit calcium overload, thereby inhibiting oxidative stress-induced apoptosis and cytoskeletal damage in adriamycin-injured rat heart cells [80]. Coronary heart disease (CHD) is another predominant cardiovascular disease and remains a serious public health burden [75]. Endothelium-dependent coronary artery vascular reactivity is a key indicator of vascular function. Endothelial dysfunction is characterized by reduced nitric oxide (NO) bioavailability [81]. In a previous study, the effects of AR extracts and their main compounds on mitochondrial bioenergetics were evaluated. The results showed that AR aqueous extracts inhibited ROS production in cardiomyocytes under oxidative stress and had the strongest antioxidant activity and protective effect. This protective effect is thought to be mediated by increases in the spare respiratory capacity and mitochondrial ATP production in stressed cells [82]. In addition, metalloproteinases are closely associated with cardiovascular disease caused by hyperhomocysteinemia. The results showed that total AR extract and *Astragalus* membranaceus total saponin (ASP) increased abnormal aortic NO production, increased superoxide dismutase (SOD) activity, decreased the concentration of the metalloproteinases MMP-2 and MMP-9, and significantly reduced abnormal aortic NO production, thereby improving vascular endothelial dysfunction in rats with hyperhomocysteinemia-induced cardiovascular disease [83].

### 5.6. Anti-Diabetic Activity

Type 1 diabetes mellitus (T1DM) and type 2 diabetes mellitus (T2DM) are widely prevalent metabolic diseases with different types of pathology. T1DM manifests itself as a result of the autoimmune destruction of pancreatic beta cells, leading to reduced insulin secretion [84]. T2DM originates from a state of insulin resistance, leading to hyperglycaemia and reduced beta-cell volume. Both diseases can lead to serious health consequences [85]. Numerous studies have demonstrated the effective biological effects of AR on T1DM and T2DM.

#### 5.6.1. Prevention and Treatment of T1DM

T1DM is an autoimmune disease which is characterized by the destruction of insulin-producing beta cells in the pancreas and a complete deficiency in insulin secretion [86]. The T lymphocyte (T cell)-mediated immune response plays an important role in the pathogenesis of T1DM. In clinical practice, *Astragalus* polysaccharide (AP) is extensively used as an immunosuppressant. There is substantial evidence that *Astragalus* polysaccharide downregulates blood glucose levels, upregulates serum insulin concentrations, increases β-cell volume and decreases the percentage of apoptotic β-cells in streptozotocin-induced T1DM mice, leading to a downregulation of the Th1/Th2 cytokine ratio and upregulation of PPARγ gene expression [87,88]. In addition, previous studies have demonstrated the preventive effect of *Astragalus* polysaccharides on T1DM by correcting the imbalance between Th1/Th2 cytokines in non-obese diabetic (NOD) mice [89]. In addition, galectin-1 (gal-1) is known to be closely associated with T cell activation and apoptosis. The finding that APS can regulate the expression of galectin-1 muscle in in vitro experiments using T1DM rats, leading to the apoptosis of CD8+ T cells, may be an important, as this is the mechanism through which AP protects pancreatic islets from β-cell CD8+ T cells in inducing apoptosis in T1DM in vivo [90].

#### 5.6.2. Prevention and Treatment of T2DM

It has been well documented that improving insulin tolerance is the main mechanism of AR and that improving insulin tolerance is the active ingredient in AR for T2DM [84]. Zhang et al. found that AP not only activated the AMPK signaling pathway to promote GLUT4 expression and glucose metabolism, but also activated the insulin signaling pathway to promote GLUT4 from intracellular vesicles to the cell membrane, thereby enhancing insulin sensitivity in mouse 3T3-L1 preadipocytes [91]. In T2DM db/db mice, Chen et al. observed that AR water extracts prevented the development of diabetes and improved renal function, and that AR water extracts modulating the IRS1-PI3K-GLUT signaling pathway significantly improved diabetic nephropathy [92]. Chen et al.’s study has demonstrated that the activation of PPARα target genes involved in db/db diabetic heart and myosin heavy chain-PPARα myocardial fatty acid uptake and oxidation is antagonized by *Astragalus* polysaccharides in db/db diabetic mice [93]. Li et al. systematically reviewed randomized and semi-randomized controlled trials (meta-analysis) to determine the role of *Astragalus* in the clinical management of DN. The results of the analysis showed that *Astragalus* injection had stronger renoprotective effects (Blood urea nitrogen (BUN), Serum creatinine (SCr), and Creatinine clearance (CCr, urinary protein) and improved systemic status (serum albumin levels) in DN patients compared to controls [94].

The normal functioning of the intestinal flora in the human body maintains a dynamic balance between the intestinal microorganisms and the organism, and is closely related to the normal functioning of the host physiology [95]. If this balance is disturbed, changes and disturbances in the intestinal flora will lead to various metabolic changes in the body, especially for the development of diabetes [95]. AR has been shown to be effective in regulating the intestinal flora in the body and in treating type 2 diabetes by improving insulin resistance and lowering blood glucose. According to traditional Chinese medicine, the most important etiological mechanism in T2DM is “Qi and Yin deficiency, heat and toxicity”, which leads to disorders of glucose metabolism, insulin resistance and damage to the intestinal mucosa, all of which can be classified as “heat” and “toxicity” in Chinese medicine [96]. The disorders of glucose metabolism, insulin resistance and intestinal mucous membrane damage can be classified as “heat” and “toxicity” in Chinese medicine. However, the role of the “middle Jiao” in the body includes receiving, decomposing, digesting and absorbing, and the blood is purified so that “evil Qi” can be removed. Wang et al. found that gavage of diabetic rats fed high-fat with the ginseng and *Astragalus* formula (*Astragalus*, ginseng, smallpox powder, yam, and raw licorice) reduced blood glucose levels, increased the expression of GLP-1 in the serum of the rats, increased the ratio of the *Bacteroidetes/Fimicutes*, and increased the relative abundance of *Bacteroides*, *Clostridium butyricum*, and *Rothia Geory and Brown* [96]. Yao et al. used “*Astragalus* turtle shell soup, ATSS (Huangqi Biejiatang, In Chinese) 黄芪鳖甲汤” to treat T2DM patients and found that the relative abundance of *Brucella*.spp decreased and that of *Bacteroides* increased after treatment, indicating that ATSS plays a positive role in regulating the dynamic balance of intestinal flora [97].

## 6. Standardization of AR

There are at least three levels of standards regarding the standardization of AR, namely efficacy standards, bioactivity standards, and production standards. AR is complex in terms of its origin, substance base, preparation process and quality control indicators, and it is for this reason that there is a lack of established standards for AR, but rather applicable standards must be developed in conjunction with the specific characteristics of specific species [98].

### 6.1. Efficacy Standards

Efficacy criteria are the first standard to be considered for AR, especially clinical efficacy. This is determined by its essential properties and cannot be considered in a single, incomplete way by considering its chemical composition, bioavailability, quality consistency and other modern drug indicators. Of course, efficacy criteria need to be measured through many clinical applications and medical observations, and for this reason, a reasonable evaluation system must be established. Unfortunately, there are few clinical studies on AR, data from clinical samples are lacking, and more studies are only to the extent of cellular or animal experiments.

### 6.2. Bioactivity Standards

The second criterion is the activity criterion, which is considered based on the physical and chemical properties of the AR (such as AR’s main ingredient, chemical ingredient contents, etc.). It is well known that it is not valid to talk about activity in isolation from dosage. Therefore, the relationships between dose and activity and between structure and activity need to be rigorously studied in order to establish AR activity criteria (such as antioxidant, anti-inflammatory, antitumor, antiviral activities, etc.). However, what is even more important for AR is batch consistency, as an extract of a natural product, so batch consistency is crucial and arguably a prerequisite for efficacy and safety. Unfortunately, different extraction and isolation methods can have a very significant impact on AR activity, and this poses a major challenge for the development of activity standards.

### 6.3. Production Standards

The production standards mentioned here cover both the cultivation and production of AR. There are no specific standards for AR cultivation and there is a lack of in-depth research. They are generally based on basic standards such as GAP (Good Agricultural Practice) and GMP (Good Manufacturing Practice). However, for the future industrialization of AR, more in-depth research is needed for the development of standard production. For example, how does the climate and soil of different production areas affect the accumulation of AR ingredients, and how do the many processes of extraction, alcohol precipitation, extraction and chromatography affect the migration and transformation of target ingredients in the AR production process? Studying the migration and transformation of the target components during the extraction, alcohol deposition, extraction, and chromatography processes is required.

## 7. Conclusions

This paper reviews the chemical composition, extraction methods and chemical analysis and biological activity of AR, a highly valued herb that is both medicinal and edible, and therefore phytochemical and pharmacological studies of AR are becoming increasingly attractive. Although research on AR has been conducted for a long time and an increasing number of interesting compounds have been isolated and identified based on the development of new extraction methods and analytical methods, many problems with the application of AR in clinical treatment are still evident due to its complex chemical composition and poor quality control. Further research is needed to clarify the compounds of AR and to standardize their quality control with the help of extraction methods and analytical methods, which may lay the foundation for the further development of traditional Chinese medicine. Regarding the standardizations of AR, there are at least three levels of criteria, namely potency criteria, bioactivity criteria and production criteria, which still need further researched in depth.

## Authors Contributions

All authors contributed equally to this research. All authors have read and agreed to the published version of the manuscript.

## Figures and Tables

**Figure 1 molecules-27-01058-f001:**
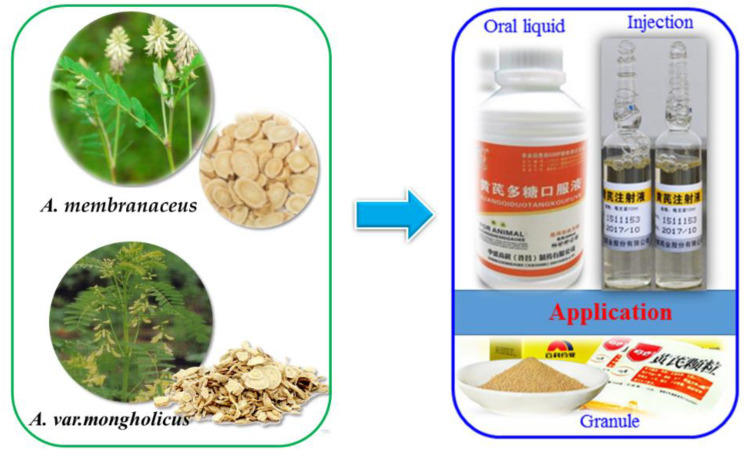
*A. membranaceus* and *A. var. mongholicu* and their commercial application.

**Figure 2 molecules-27-01058-f002:**
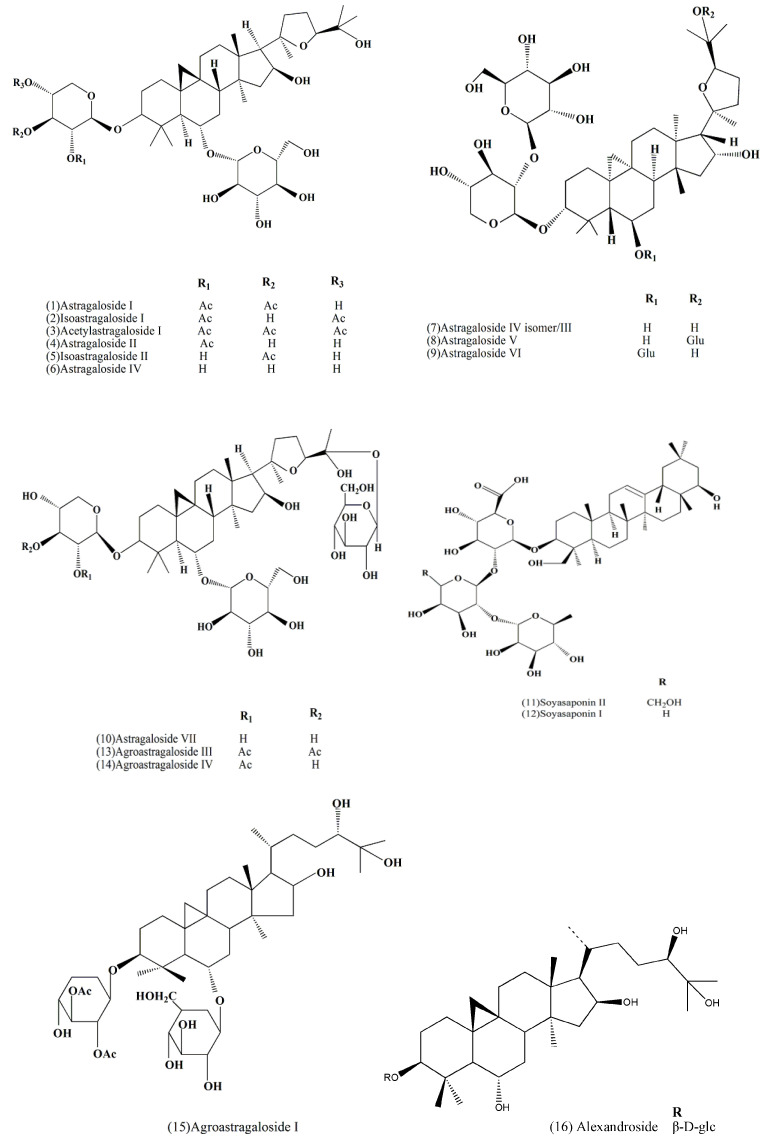
Chemical structures of major saponins identified from AR.

**Figure 3 molecules-27-01058-f003:**
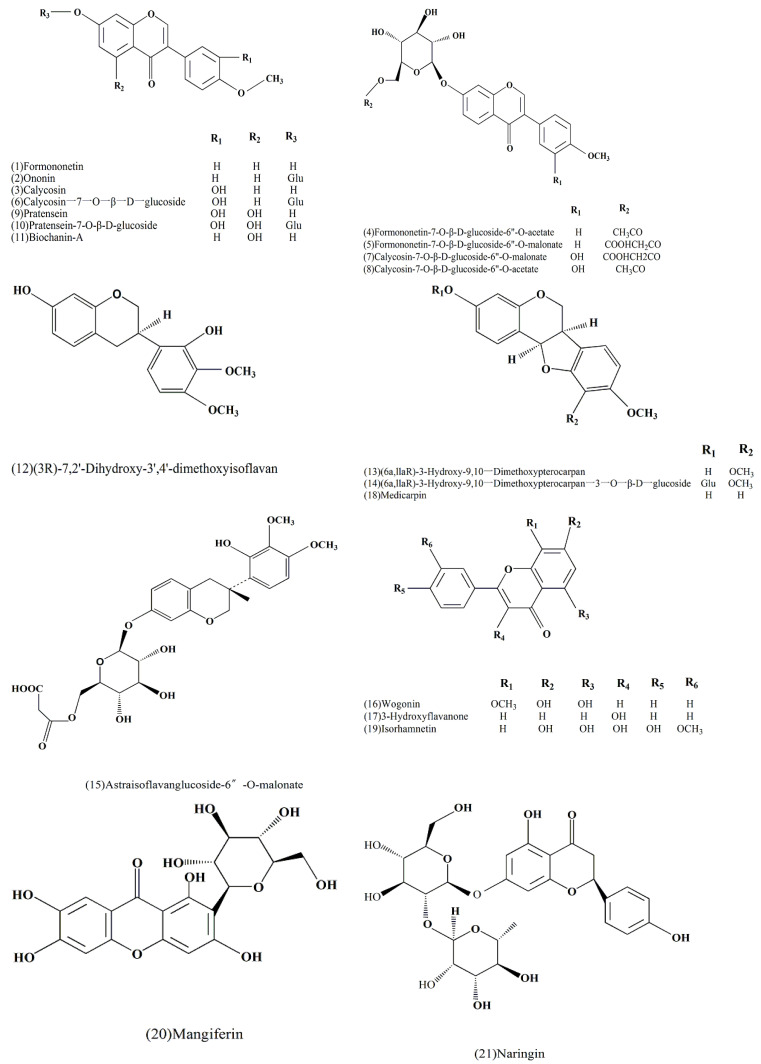
Chemical structures of major flavonoids.

**Figure 4 molecules-27-01058-f004:**
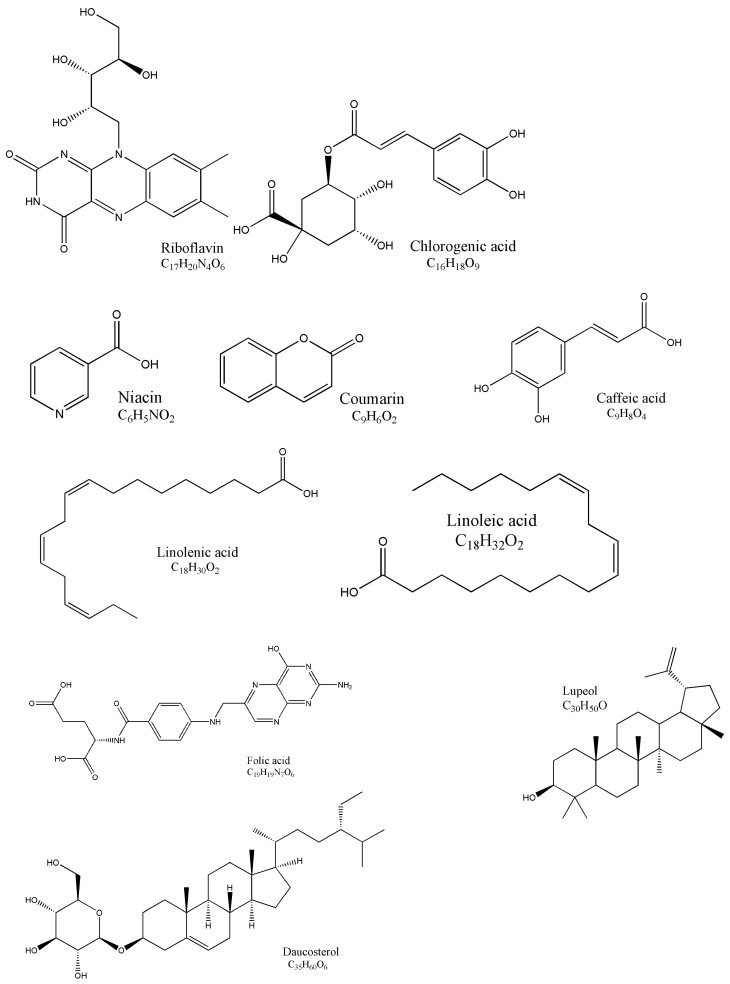
Chemical structures of the other 10 active components identified from AR.

**Figure 5 molecules-27-01058-f005:**
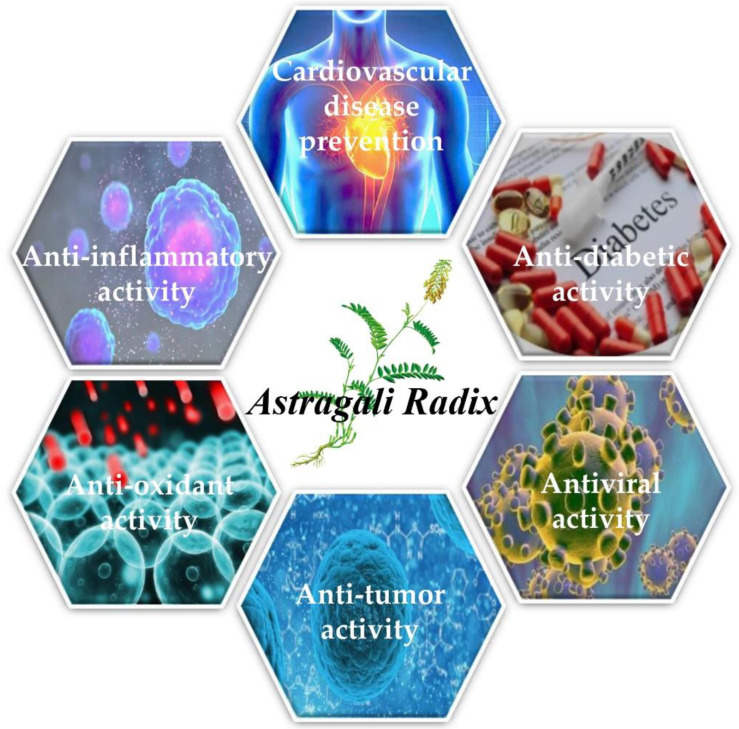
Biological activities of AR.

**Figure 6 molecules-27-01058-f006:**
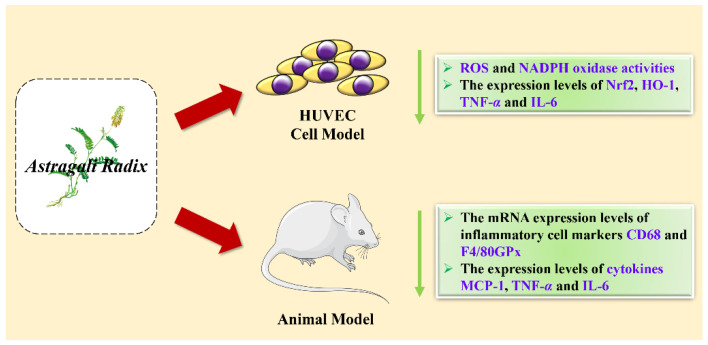
The anti-inflammatory activity mechanism of AR in cell and animal models.

**Figure 7 molecules-27-01058-f007:**
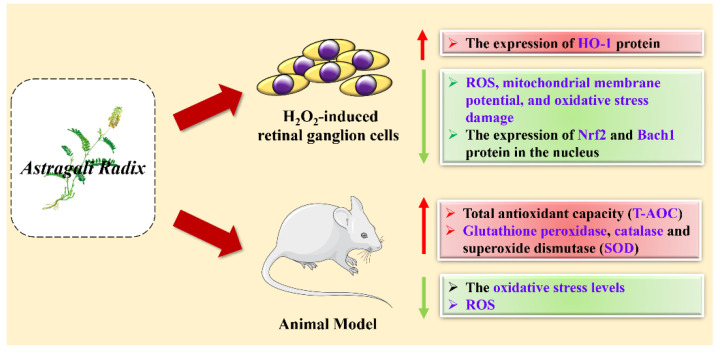
The antioxidant activity mechanism of AR in cell and animal models.

**Figure 8 molecules-27-01058-f008:**
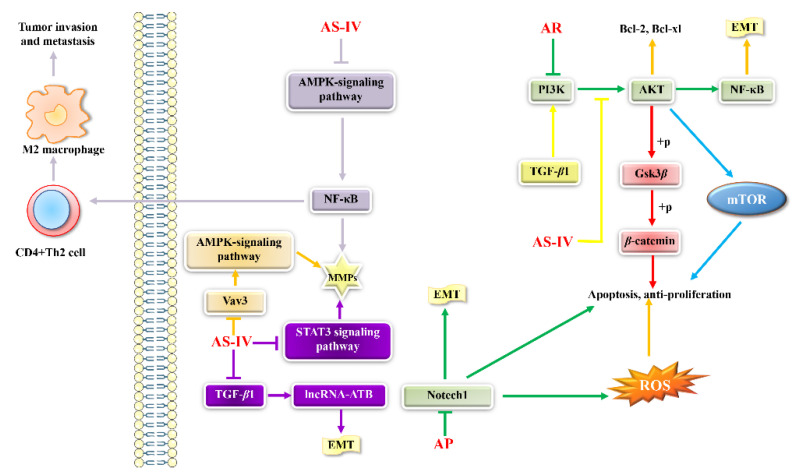
Several mechanisms linking AR treatment of cancer. Arrows and bar-headed lines represent signaling activation and inhibition, respectively [53].

**Table 1 molecules-27-01058-t001:** Information regarding the polysaccharides identified from AR.

No.	Name	Molecular Weight (Da)	Identification Method	Classification	Structure	Source	Refs.
1	AG-1	-	400 M NMR	Glucan	-	*A. membranaceus*	[13]
2	AG-2	-	400 M NMR	Glucan	-	*A. membranaceus*	[13]
3	AH-1	-	400 M NMR	Heteropolysaccharide	-	*A. membranaceus*	[13]
4	AH-2	-	400 M NMR	Heteropolysaccharide	-	*A. membranaceus*	[13]
5	APS-I	1.7 × 10^6^	HPLC (C18), TLC	Heteropolysaccharide	-	*A. membranaceus*	[7]
6	APS-II	1.2 × 10^6^	HPLC (C18), TLC	D-Glucan	Dextran bonded mainly with-(1 → 4)-d-glycosidic linkage	*A. membranaceus*	[7]
7	APS-III	3.5 × 10^4^	400 M NMR	D-Glucan	Dextran bonded mainly with -(1 → 4)-d-glycosidic linkage	*A. membranaceus*	[14]

**Table 2 molecules-27-01058-t002:** Detailed information of major saponins identified from AR.

No.	Compound Name	Molecular Formula	Molecular Weight	ESI–MS		APCI–MS		Source	Refs.
				Parent Ion (*m*/*z*)	Fragment Ion (*m*/*z*)	Parent Ion (*m*/*z*)	Fragment Ion (*m*/*z*)		
1	Astragaloside I	C_45_H_72_O_16_	869.04	-	-	867.7 [M − H]^−^	807.5 [M − H-Ac]^−^	*A. membranaceus*	[15]
2	Isoastragaloside I	C_45_H_72_O_16_	869.04	-	-	-	-	*A. membranaceus*	[21]
3	Acetylastragaloside I	C_47_H_74_O_17_	911.08	-	-	-	-	*A. membranaceus*	[21]
4	Astragaloside II	C_43_H_70_O_15_	827.00	-	-	825.7 [M − H]^−^	765.5 [M − H-Ac]^−^	*A. membranaceus*	[15]
5	Isoastragaloside II	C_43_H_70_O_15_	827.00	-	-	-	-	*A. membranaceus*	[21]
6	Astragaloside IV	C_41_H_68_O_14_	784.97	-	-	783.7 [M − H]^−^	651.4 [M − H−(Xyl-H_2_O)]^−^	*A. membranaceus*	[15]
7	Astragaloside IV isomer/III	C_41_H_68_O_14_	784.97	-	-	783.7 [M − H]^−^	651.7 [M − H-(Xyl-H_2_O)]^−^	*A. membranaceus*	[15]
8,9,10	Astragaloside V/VI/VII	C_47_H_78_O_19_	947.11	-	-	945.6 [M − H]^−^	783.6 [M − H-(Glu-H_2_O)]^−^	*A. membranaceus*	[15]
11	Soyasaponin II	C_47_H_76_O_17_	913.10	914 [M + H]^+^	457.4	-	-	*A. membranaceus*	[22]
12	Soyasaponin I	C_48_H_78_O_18_	943.12	944 [M + H]^+^	617.4	-	-	*A. membranaceus*	[22]
13	Agroastragaloside III	C_51_H_82_O_21_	1031.18	-	-	-	-	*A. membranaceus*	[20]
14	Agroastragaloside IV	C_49_H_80_O_20_	989.14	-	-	-	-	*A. membranaceus*	[20]
15	Agroastragaloside I	C_45_H_74_O_16_	871.06	-	-	-	-	*A. membranaceus*	[17]
16	Alexandroside	C36H62O10	654.87	-	-	-	-	*A. membranaceus A. membranaceus*	[20]

**Table 3 molecules-27-01058-t003:** Detailed information of major flavonoids identified from AR.

No.	Compounds Name	Molecular Formula	Molecular Weight	ESI–MS		APCI–MS		Source	Refs.
				Parent Ion (*m*/*z*) [M + H]^+^/[M + Na]^+^	Fragment Ion (*m*/*z*)	Parent Ion (*m*/*z*) [M + H]^+^	Fragment Ion (*m*/*z*)		
1	Formononetin	C_16_H_12_O_4_	268.27	269	254	-	-	both	[20,24]
2	Ononin	C_22_H_22_O_9_	430.40	431	269	-	-	both	[20,24]
3	Calycosin	C_16_H_12_O_5_	284.26	285	275	-	-	both	[20,24]
4	formononetin-7-*O*-β-d-glucoside-6”-*O*-acetate (4), (8),	C_24_H_24_O_10_	472.44	-	-	473	269	both	[23]
5	Formononetin-7-*O*-β-d-glucoside-6”-*O*-malonate	C_25_H_24_O_12_	516.45	517	269	-	-	*A. membranaceus*	[24]
6	Calycosin-7-*O*-β-d-glycoside	C_22_H_22_O_10_	446.40	447	285	-	-	both	[24]
7	Calycosin-7-*O*-β-d-glucoside-6”-*O*-malonate	C_25_H_24_O_13_	532.45	533	285	-	-	*A. membranaceus*	[24]
8	Calycosin-7-*O*-β-d-glucoside-6”-*O*-acetate	C_24_H_24_O_11_	488.44	-	-	489	285	both	[23]
9	Pratensein	C_16_H_12_O_6_	300.26	-	-	301	269	both	[23]
10	Pratensein-7-*O*-β-d-glycoside	C_22_H_22_O_11_	462.40	-	-	463	301	both	[23]
11	Biochanin-A	C_16_H_12_O_5_	284.30	-	-	285	-	*A. membranaceus*	[23]
12	(3R)-7,2′-Dihydroxy-3′,4′-dimethoxyisoflavan	C_17_H_18_O_5_	302.32	-	-	303	167	both	[23,24]
13	(6a,llaR)-3-Hydroxy-9,10-Dimethoxypterocarpan	C_17_H_16_O_5_	300.31	301	152	-	-	*A. membranaceus*	[23,24]
14	(6a,llaR)-3-Hydroxy-9,10-Dimethoxypterocarpan-3-*O*-β-d-glycoside	C_23_H_26_O_10_	462.15	463	301	-	-	*A. membranaceus*	[24]
15	Astraisoflavanglucoside-6”-*O*-malonate	C_26_H_30_O_13_	550	551	303	-	-	*A. membranaceus*	[24]
16	Wogonin	C_16_H_12_O_5_	284.27	285.1	270.2	-	-	*A. membranaceus*	[22]
17	3-Hydroxyflavanone	C_15_H_12_O_3_	240.25	241	-	-	-	*A. membranaceus*	[22]
18	Medicarpin	C_16_H_14_O_4_	270.28	271	-	-	-	*A. membranaceus*	[22]
19	Isorhamnetin	C_16_H_12_O_7_	316.26	317	-	-	-	*A. membranaceus*	[22]
20	Mangiferin	C_19_H_18_O_11_	422.34	423	-	-	-	*A. membranaceus*	[22]
21	Naringin	C_27_H_32_O_14_	580.53	603	-	-	-	*A. membranaceus*	[22]

**Table 4 molecules-27-01058-t004:** Advantages and disadvantages of extraction methods for AR studies.

	Advantages	Disadvantages
Water extraction	Simple operation and low production costs (maximal extraction rate of AP was 16.32%)	Large energy consumption and low extraction rate
Ethanol Reflux extraction	Wide range of applications, simple equipment, good extraction effect	Long extraction time, solvent residue
Microwave-assisted Extraction	Penetrating heating, time saving, high efficiency, energy saving (maximal extraction rate of AP was 32%, 49% improvement compared to conventional water extraction)	Volatile components gradually dissipate as the extraction time increases.
Ultrasonic extraction	Simple operation, high efficiency, time saving and energy saving (maximal extraction rate of AP was 30.28%, 46.23% improvement compared to conventional water extraction)	Sound pollution
Enzyme extraction	High specificity and efficiency (maximal extraction rate of AP was 29.96%, 45.52% improvement compared to conventional water extraction)	High production costs

## Data Availability

Not applicable.

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
