# Peer review of "Advances in Chemical Composition, Extraction Techniques, Analytical Methods, and Biological Activity of Astragali Radix"

_molecules, 2022, doi:10.3390/molecules27031058_

Round 1

Reviewer 1 Report

  1. Page 1. 1. Introduction. Authors pointed out the need for standardization of Astragalus. But in Page 18. 5. Conclusion. Authors did not answer the question. Please add the description or suggestion for the standardization of Astragals.
  2. Page 3. Table 1. Is it possible to provide the structures for AG, AH, and APS? In Identification method, I would suggest to add the condition, such as NMR ? MHz, and the column for HPLC.
  3. All the structure formats and the font style should be the same.
  4. Page 14. Table 4. Please add the increased % compared to traditional extraction method.
  5. Page 15. 4.2 Chromatographic Analysis. What are the 8 flavonoids? Please list the name.
  6. Page 17. 5.3. Anti-tumor activity. What AI stand for ?

Author Response

  1. Page 1. 1. Introduction. Authors pointed out the need for standardization of Astragalus. But in Page 18. 5. Conclusion. Authors did not answer the question. Please add the description or suggestion for the standardization of Astragals.

Thank you for your comments. At the conclusion about the standardization of Astragalus was supplemented and is presented separately in Part 6 by us. The details are as follows.

Page 20~21, Line 556~593,

  1. Standardization of AR

There are at least 3 levels of standards regarding the standardization of AR, namely efficacy standards, bioactivity standards, and production standards. AR is complex in origin, substance base, preparation process and quality control indicators, and it is for this reason that there is a lack of established standards for AR, but rather applicable standards must be developed in conjunction with the specific characteristics of specific species.

6.1. Efficacy standards

Efficacy criteria are the first standard to be considered for AR, especially in the clinical efficacy. This is determined by its essential properties and cannot be considered in a single, incomplete way by considering its chemical composition content, bioavailability, quality consistency and other modern drug indicators. Of course, efficacy criteria need to be measured through many clinical applications and medical observations, and for this reason, a reasonable evaluation system must be established. Unfortunately, there are few clinical studies on AR, data from clinical samples are lacking, and more studies are only to the extent of cellular or animal experiments.

6.2. Bioactivity standards

The second criterion is the activity criterion, which is considered based on the physical and chemical properties of the AR (such as AR main ingredient content, chemical ingredient content, etc.). It is well known that it is not valid to talk about activity in isolation from dosage. Therefore, the relationship between dose and activity, and between structure and activity, needs to be rigorously studied in order to establish AR activity criteria (such as antioxidant, anti-inflammatory, anti-tumors, anti-viral activity etc.). But what is even more important for AR is batch consistency, as an extract of a natural product, so batch consistency is crucial and arguably a prerequisite for efficacy and safety. Unfortunately, different extraction and isolation methods can have a very significant impact on AR activity, and this poses a major challenge for the development of activity standards.

6.3. Production standards

The production standards mentioned here cover both the cultivation and production of AR. There are no specific standards for AR cultivation and there is a lack of in-depth research. They are generally based on basic standards such as GAP (Good Agricultural Practice) and GMP (Good Manufacturing Practice). But for the future industrialization of AR, more in-depth research is needed for the development of standard production. For example, how does the climate and soil of different production areas affect the accumulation of AR ingredients, and how do the many processes of extraction, alcohol precipitation, extraction and chromatography affect the migration and transformation of target ingredients in the AR production process? The study of the migration and transformation of the target components during the extraction, alcohol deposition, extraction, and chromatography processes.

Page 21, Line 606~608, Regarding the standardizations of AR, there are at least 3 levels of criteria, namely potency criteria, bioactivity criteria and production criteria, which still need further in-depth re-search.

  1. Page 3. Table 1. Is it possible to provide the structures for AG, AH, and APS? In Identification method, I would suggest to add the condition, such as NMR ? MHz, and the column for HPLC.

Thank you for your comments. The references were re-read on their own and specific details of NMR and HPLC have been added to the tables. Unfortunately, the references cited do not contain detailed structural reports on the AG, AH polysaccharide.

  1. All the structure formats and the font style should be the same.

Thank you for your comments. We have redrawn the structural style and standardized the font size and typeface.

  1. Page 14. Table 4. Please add the increased % compared to traditional extraction method.

Thank you for your comments. We re-searched the literature and compared the different extraction methods with traditional water extraction for the extraction rate of the main component of Astragalus polysaccharide (AP) and calculated the growth rate.

  1. Page 15. 4.2 Chromatographic Analysis. What are the 8 flavonoids? Please list the name.

Due to an oversight on our side, this was not explained clearly. We have re-searched the literature and added the relevant 8 flavonoids at the original text. The details are as follows.

Page 14, Line 307~309, Eight flavonoids (9,10-dimethoxypterocarpan 3-O-β-D-glucoside, ononin, 2′,4′-dimethoxy-3′-hydroxyisoflavan 6-O-β-D-glucoside, 10-hydroxy-3,9-dimethoxypterocarpan, afromosin, calycosin, formononetin, and rutin)

  1. Page 17. 5.3. Anti-tumor activity. What AI stand for?

Thank you for your comments. Due to an oversight on our part, this is a spelling error. The word “AI” here should be changed to “AR”.

Reviewer 2 Report

The manuscript should be accepted for publication. I have just a few minor comments/suggestions for authors that I have highlighted in the attached pdf.

Author Response

Reviewer #2:

  1. Astragalus membranaceus” please add a family.

Thank you for your comments. We have added the family name of Astragalus membranaceus. The details are as follows.

Page 1, Line 26~27, Astragalus membranaceus (Fisch.) Bge. or Astragalus membranaceus (Fisch.) Bge. var.mongholicus (Bge.) Hsiao (Fabaceae)

  1. In Fig.1, without dot and erase dot on the image.

It is our negligence that we failed to explain clearly. Figure. 1 was redrawn by us and supplemented with a diagram of A. var. mongholicu.

  1. In 2.2, “sponins” should be changed to “saponins”.

Thank you for your comments. Due to an oversight on our part, this is a spelling error. The word “Sponins” here should be changed to “Saponins”.

  1. In 3.1, “Astragalus membranaceus” should be italic, and check in whole manuscript, somewhere is italic somewhere not, please make it uniform.

Thank you for your comments. We have carefully checked the full text and amended what should be italicized.

  1. The abbreviation AP should be introduced at the place where the astragalus polysaccharides are first mentioned and then further in the text use only the abbreviation.

Thank you for your comments. We have made the abbreviation AP should be written in full at the first mention of Astragalus polysaccharide and then continue to use that abbreviation throughout the text.

  1. Relevant references are missing from the text, please add the corresponding references in the appropriate places.

Thank you for your comments. We have carefully checked the references and added the relevant references where they are missing from the text.

Reviewer 3 Report

This paper reviews the extraction techniques, chemical composition, analytical methods, and biological activities of Astragali Radix. It also provides theoretical support for better development and utilization of AR. However, this paper seems to lack innovation, does not provide the authors' thoughts and comments, and is more like mere data compilation. In addition, the same topic has been recently summarized by Zhang, Chun-hong, et al. ("Ethnopharmacology, phytochemistry, pharmacology, toxicology and clinical applications of Radix Astragali." Chinese journal of integrative medicine (2019): 1-12.), and  Guo, Zhenzhen, et al. ("A systematic review of phytochemistry, pharmacology and pharmacokinetics on Astragali Radix: Implications for Astragali Radix as a personalized medicine." International journal of molecular sciences 20.6 (2019): 1463.). Therefore, I suggest that the article does not meet the criteria for publication in the molecules journal. The specific comments are as follows for the authors' reference.

  1. It is recommended that the plural of "advances" be used in the title. Only the first word in the title is capitalized or all words except prepositions are capitalized, please be consistent. In the manuscript, the chemical composition is introduced first, followed by the extraction technique, so the title should preferably be in the same order as the article.

  1. There are many tense and grammatical errors in the manuscript, which need to be revised and polished.

  1. It is suggested to remove "Traditional Chinese medicine" from the "Keywords" and change "chemical analysis" to "analytical methods".

  1. The 2020 edition of the Chinese Pharmacopoeia has been published, and the relevant contents of the text should be based on the latest edition.

  1. AR has been used to represent "Astragali Radix" in the article. Why did many "Astragalus" appear in the first paragraph of "Introduction".

  1. There are two sources of AR. Why was only one of them shown in Figure 1?

  1. In Tables 2 and 3, the column "Fragment ions", [M+H]+ is not a fragment ion.

  1. The font size varies in the chemical structure diagrams.

  1. In section 3.1.1, there should be a space between the number and the unit. In addition, please unify the number of decimal places, such as 1 h and 1.0 h.

  1. It is suggested that heading 4, "Chemical analysis", be changed to "Analytical methods". In addition, in this section, the manuscript is too homogeneous in terms of the examples cited. In fact, there is much more than these reports in the field and the authors could have used a table to summarize and organize them.

  1. At the end of section 4.2, the reference is missing.

  1. There are also some abbreviations that appear for the first time in the manuscript, but the full names are not given.

  1. The content of the review is too superficial, which looks uninspiring when compared with the published articles, for example, pharmacological effects rarely involve mechanisms. Since the "Introduction" mentioned the different preparations of AR, is it possible to add a part of clinical application to the main text?

  1. Some of the literature in the manuscript is outmoded, and it would be better to cite articles published in recent years. In addition, as a review on AR, a common traditional Chinese medicine, there are too few recent references.

  1. The manuscript lacks in-depth discussion and prospects.

  1. The format of references is not uniform.

Author Response

Reviewer #3:

  1. It is recommended that the plural of "advances" be used in the title. Only the first word in the title is capitalized or all words except prepositions are capitalized, please be consistent. In the manuscript, the chemical composition is introduced first, followed by the extraction technique, so the title should preferably be in the same order as the article.

Thank you for your comments. We have rewritten the title to comply with the spelling requirements of the title. The details are as follows.

Page 1, Line 2~3, changed “Advance in Extraction techniques, chemical composition, analytical methods, and biological activity of Astragali Radix” to “Advances in Chemical Composition, Extraction Techniques, Analytical Methods, and Biological Activity of Astragali Radix

  1. There are many tense and grammatical errors in the manuscript, which need to be revised and polished.

It is our negligence that we failed to explain clearly. We invited a professional native-speaking editor to do the linguistic touch-ups.

  1. It is suggested to remove “Traditional Chinese medicine” from the “Keywords” and change “chemical analysis” to “analytical methods”.

Thank you for your comments. The “Traditional Chinese medicine” from the “Keywords” was removed and “chemical analysis” was changed to “analytical methods”. The details are as follows.

Page 1, Line 20~21, change “Keywords: Astragali Radix; Traditional Chinese medicine; Extraction technologies; Chemical constituents; Chemical analysis; Biological activity” to “Keywords: Astragali Radix; Extraction technologies; Chemical constituents; Analytical methods; Biological activity”.

  1. The 2020 edition of the Chinese Pharmacopoeia has been published, and the relevant contents of the text should be based on the latest edition.

Thank you for your comments. The latest edition (2020 version) of the Chinese Pharmacopoeia was retrieved and updated by us with the description of Astragalus in the paper. Details are given below.

Page 1, Line 26~28, change to “According to the Pharmacopoeia of the People’s Republic of China (2020 edition), AR is defined as the dried root of Astragalus membranaceus (Fisch.) Bge. or Astragalus membranaceus (Fisch.) Bge. var.mongholicus (Bge.) Hsiao (Fabaceae)”

  1. AR has been used to represent “Astragali Radix” in the article. Why did many “Astragalus” appear in the first paragraph of "Introduction".

Thank you for your comments. We have replaced the word “Astragalus” with “AR” in the "Introduction".

  1. There are two sources of AR. Why was only one of them shown in Figure 1?

It is our negligence that we failed to explain clearly. Figure. 1 was redrawn by us and supplemented with a diagram of A. var. mongholicu

  1. In Tables 2 and 3, the column "Fragment ions", [M+H]+ is not a fragment ion.

Thank you for your comments. The errors in Tables 2 and 3 were modified by us. 

  1. The font size varies in the chemical structure diagrams.

Thank you for your comments. We have redrawn the structural style and standardized the font size and typeface.

  1. In section 3.1.1, there should be a space between the number and the unit. In addition, please unify the number of decimal places, such as 1 h and 1.0 h.

Thank you for your comments. We have carefully checked all figures and their units throughout the text. Details of the changes are given below.

Page 11~12, Line 157~166, changed to “Applying statistics-based experimental designs to optimize the extraction process of Astragalus Polysaccharides (AP). The optimal conditions were: extraction time 3 h, liquid-to-solid ratio 4, particle size 33.8 mesh. Under these conditions, the maximal extraction rate of AP was 16.32% [23]. The water reflux extraction technology of Astragalus polysaccharides was optimized, the ratio of solid to liquid, extraction time, extraction times and immersion duration were selected to be experimental factors, the content of Astragalus polysaccharides was used as the inspection index. The optimum extraction process conditions were: ratio of solid to liquid 1:12, extraction time 1 h, immersion duration 1 h, extraction times 3 times. Under these conditions, Astragalus polysaccharides accounted for 6.4% of the original herbs [24].”

  1. It is suggested that heading 4, "Chemical analysis", be changed to "Analytical methods". In addition, in this section, the manuscript is too homogeneous in terms of the examples cited. In fact, there is much more than these reports in the field and the authors could have used a table to summarize and organize them.

Thank you for your comments. “Chemical analysis” have been changed to "Analytical methods".

In page 15, change to “. A thin-layer chromatography (TLC) method for determination of astragalin in AR was established. Using silica gel G thin board. Chloroform-methyl alcohol-water (13:6:2) was used as developer. The detective wavelength and reference wavelength were at 530 nm and 700 nm, respectively. Results indicated that the linear relationship of astragalin curves was good. The average recovery rate was 98.72% with RSD=3.5% [22].”

  1. At the end of section 4.2, the reference is missing.

Thank you for your comments. References have been added to the end of section 4.2. The details are as follows.

Page 15, Line 314~319. Changed to “A thin-layer chromatography (TLC) method for determination of astragalin in AR was established. Using silica gel G thin board. Chloroform-methyl alcohol-water (13:6:2) was used as developer. The detective wavelength and reference wavelength were at 530 nm and 700 nm, respectively. Results indicated that the linear relationship of astragalin curves was good. The average recovery rate was 98.72% with RSD=3.5%.[28]

  1. There are also some abbreviations that appear for the first time in the manuscript, but the full names are not given.

Thank you for your comments. Some of the abbreviations appear for the first time in the manuscript and the full names have been re-added by us. The details are as follows.

Page 13, Line 239, added to “Microwave-assisted Extraction (MAE)”.

Page 14, Line 293~294, added to “Inductively coupled plasma atomic emission spectrometry (ICP-AES)”.

Page 14, Line 298, added to “Surface enhanced Raman scattering (SERS)”.

Page 15, Line 313~314, added to “thin-layer chromatography (TLC)”.

Page 16. Line 361, added to “oxidase low density lipoprotein (OX-LDL)”.

Page 17, Line 399, added to “Streptozocin (STZ)”

Page 17, Line 403, added to “total antioxidant capacity (T-AOC)”

Page 21, Line 530~533, added to “The results of the analysis showed that Astragalus injection had stronger renoprotective effects (Blood urea nitrogen (BUN), Serum creatinine (SCr), Creatinine clearance (CCr, uri-nary protein) and improved systemic status (serum albumin levels) in DN patients com-pared to controls [94].”

  1. The content of the review is too superficial, which looks uninspiring when compared with the published articles, for example, pharmacological effects rarely involve mechanisms. Since the “Introduction” mentioned the different preparations of AR, is it possible to add a part of clinical application to the main text?

Thank you for your comments. The pharmacological effects and their associated mechanisms of action (at the cellular and animal levels) are re-addressed and the pharmacological activity of AR is summarized and discussed in the context of relevant clinical data and Meta-analysis results. It is noteworthy that AR, as a major component of some ancient Chinese herbal formulas, has been added to the study of its regulation of intestinal microorganisms and its Chinese herbal theory of "Qi".

Page 17, Line 382~386, added to “Antioxidants have many positive effects on human health, protecting the body and cells from ROS and free radical damage and preventing the oxidation of biological macromolecules such as DNA, membrane lipids and proteins [7]. As a result, over the last few years there has been an increasing interest in developing safer and more effective natural antioxidants to prevent oxidative damage in the food and pharmaceutical industries [7].”

Page 17, Line 387~388, added to “The antioxidant activity of AR has been extensively studied. The antioxidant activity mechanisms of AR are presented in Figure. 7.”

Page 19~20, Line 464~490, added

5.5. Cardiovascular disease prevention

An increasing number of experiments have confirmed the effectiveness of AR in inhibiting cardiovascular diseases such as myocardial ischemia-reperfusion injury [72], myocardial hypertrophy [73], vascular endothelial dysfunction [74,75], coronary artery disease [76], atherosclerosis [77], myocardial fibrosis [78], and viral myocarditis [79].

The sarcoplasmic reticulum (SR) plays an essential role in intracellular calcium regulation, and the sarcoplasmic reticulum Ca2+- ATPase (SERCA2a) in the SR is responsible for sarcoplasmic calcium reuptake. Many studies have shown that the expression of SERCA2a is significantly reduced after myocardial injury. AR has been reported to pro-mote SERCA2a expression, enhance calcium reuptake in the SR and inhibit calcium over-load, thereby inhibiting oxidative stress-induced apoptosis and cytoskeletal damage in adriamycin-injured rat heart cells [80]. Coronary heart disease (CHD) is another predominant cardiovascular disease and remains a serious public health burden [75]. Endothelium-dependent coronary artery vascular reactivity is a key indicator of vascular function. Endothelial dysfunction is characterised by reduced nitric oxide (NO) bioavailability [81]. In a study, the effects of AR extracts and their main compounds on mitochondrial bioenergetics were evaluated. The results showed that AR aqueous extracts inhibited ROS pro-duction in cardiomyocytes under oxidative stress and had the strongest antioxidant activity and protective effect. This protective effect is thought to be mediated by increasing the spare respiratory capacity and mitochondrial ATP production in stressed cells [82]. In addition, metalloproteinases are closely associated with cardiovascular disease caused by hyperhomocysteinemia. The results showed that total AR extract and Astragalus membranaceus total saponin (ASP) increased abnormal aortic NO production, increased superoxide dismutase (SOD) activity, decreased the concentration of metalloproteinases MMP-2 and MMP-9, and significantly reduced abnormal aortic NO production. thereby improving vascular endothelial dysfunction in rats with hyperhomocysteinemia-induced cardiovascular disease [83].”

Page 20~21, Line 491~555, added,

5.6. Anti-diabetic activity

Type 1 diabetes mellitus (T1DM) and type 2 diabetes mellitus (T2DM) are widely prevalent metabolic diseases with different types of pathology. T1DM manifests itself as a result of autoimmune destruction of pancreatic beta cells, leading to reduced insulin secretion [84]. T2DM originates from a state of insulin resistance, leading to hyperglycaemia and reduced beta-cell volume. Both diseases can lead to serious health consequences [85]. Numerous studies have demonstrated the effective biological effects of AR on T1DM and T2DM.

5.6.1. Prevention and treatment of T1DM

T1DM is an autoimmune disease which is characterized by the destruction of insulin-producing beta cells in the pancreas and a complete deficiency in insulin secretion [86]. T lymphocyte (T cell)-mediated immune response plays an important role in the pathogenesis of T1DM. In clinical practice, Astragalus polysaccharide (AP) is extensively used as an immunosuppressant. There is substantial evidence that Astragalus polysaccharide down-regulates blood glucose levels, up-regulates serum insulin concentrations, increases β-cell volume and decreases the percentage of apoptotic β-cells in streptozotocin-induced T1DM mice, leading to a down-regulation of the Th1/Th2 cytokine ratio and up-regulation of PPARγ gene expression [87,88]. In addition, studies have demonstrated the preventive effect of Astragalus polysaccharides on T1DM by correcting the imbalance between Th1/Th2 cytokines in non-obese diabetic (NOD) mice [89]. In addition, galectin-1 (gal-1) is known to be closely associated with T cell activation and apoptosis. The finding that APS can regulate the expression of galectin-1 muscle in vitro experiments in T1DM rats, leading to apoptosis of CD8+ T cells, may be an important mechanism through which AP protects pancreatic islets from β-cell CD8+ T cells in inducing apoptosis in T1DM in vivo [90].

5.6.2. Prevention and treatment of T2DM

It has been well documented that improving insulin tolerance is the main mechanism of AR and that improving insulin tolerance is the active ingredient in AR for T2DM [84]. Zhang et al. found that AP not only activated the AMPK signaling pathway to pro-mote GLUT4 expression and glucose metabolism, but also activated the insulin signaling pathway to promote GLUT4 from intracellular vesicles to the cell membrane, thereby enhancing insulin sensitivity in mouse 3T3-L1 preadipocytes [91]. In T2DM db/db mice, Chen et al. observed that AR water extracts prevented the development of diabetes and improved renal function, and that AR water extracts modulating the IRS1-PI3K-GLUT signaling pathway significantly improved diabetic nephropathy [92]. Chen et al.’s study has demonstrated that activation of PPARα target genes involved in db/db diabetic heart and myosin heavy chain-PPARα myocardial fatty acid uptake and oxidation is antagonized by astragalus polysaccharides in db/db diabetic mice [93]. Li et al. systematically re-viewed randomised and semi-randomised controlled trials (Meta-analysis) to determine the role of Astragalus in the clinical management of DN. The results of the analysis showed that Astragalus injection had stronger renoprotective effects (Blood urea nitrogen (BUN), Serum creatinine (SCr), Creatinine clearance (CCr, urinary protein) and improved systemic status (serum albumin levels) in DN patients compared to controls [94].

The normal functioning of the intestinal flora in the human body maintains a dynamic balance between the intestinal microorganisms and the organism, and is closely related to the normal functioning of the host physiology [95]. If this balance is disturbed, changes and disturbances in the intestinal flora will lead to various metabolic changes in the body, especially for the development of diabetes [95]. AR has been shown to be effective in regulating the intestinal flora in the body and in treating type 2 diabetes by im-proving insulin resistance and lowering blood glucose. According to traditional Chinese medicine, the most important etiological mechanism in T2DM is “Qi and Yin deficiency, heat and toxicity”, which leads to disorders of glucose metabolism, insulin resistance and damage to the intestinal mucosa, all of which can be classified as "heat" and "toxicity" in Chinese medicine [96]. The disorders of glucose metabolism, insulin resistance and intestinal mucous membrane damage can be classified as "heat" and "toxicity" in Chinese medicine. However, the role of the “middle Jiao” in the body includes receiving, decom-posing, digesting and absorbing, and the blood is purified so that “evil Qi” can be re-moved. Wang et al. found that gavage of diabetic rats fed high-fat with the ginseng and astragalus formula (Astragalus, ginseng, smallpox powder, yam, and raw licorice) reduced blood glucose levels, increased the expression of GLP-1 in the serum of the rats, in-creased the ratio of the Bacteroidetes/Fimicutes, and increased the relative abundance of the Bacteroides, Clostridium butyricum, and Rothia Geory and Brown [96]. Yao et al. used “Astragalus turtle shell soup, ATSS (黄芪鳖甲汤)” to treat T2DM patients and found that the relative abundance of Brucella.spp decreased and that of Bacteroides increased after treatment, indicating that ATSS plays a positive role in regulating the dynamic balance of intestinal flora [97].

  1. Some of the literature in the manuscript is outmoded, and it would be better to cite articles published in recent years. In addition, as a review on AR, a common traditional Chinese medicine, there are too few recent references.

Thank you for your comments. References from the last five years were retrieved and added to this review, and existing articles on the use of AR in clinical practice and related Meta-analyses were added to this review to supplement missing clinical data and relevant discussions. The details in the “References” are as follows.

Page 22~28, Line 615~864, Updated references,

  1. Kafle, B.; Baak, J.P.A.; Brede, C. Major bioactive chemical compounds in astragali radix samples from different vendors vary greatly. Plos One 2021, 16.1-8.
  • This paper focuses on describing the differences in the biological activities of different suppliers of Astragalus.
  1. Sun, X.-R.; Wei, G.-J.; Wang, H.-Y.; Zhao, W.-W.; Zhang, Z.; Zhang, T.; Wei, F.; Zhang, Y.-J. Evaluation of astragali radix quality grade based on appearance characteristics and internal ingredients. Zhongguo Zhong yao za zhi = Zhongguo zhongyao zazhi = China journal of Chinese materia medica 2021, 46, 966-971.
  2. Du, N.; Xu, Z.; Gao, M.; Liu, P.; Sun, B.; Cao, X. Combination of ginsenoside rg1 and astragaloside iv reduces oxidative stress and inhibits tgf-beta 1/smads signaling cascade on renal fibrosis in rats with diabetic nephropathy. Drug Design Development and Therapy 2018, 12, 3517-3524.
  3. Cheng, J.-H.; Guo, Q.; Sun, D.-W.; Han, Z. Kinetic modeling of microwave extraction of polysaccharides from astragalus membranaceus. Journal of Food Processing and Preservation 2019, 43, 1-9.
  4. Pan, C.; Wang, H.; Shan, H.; Lu, H. Preparative isolation and purification of calycosin and formononetin from astragali radix using hydrolytic extraction combined with high speed countercurrent chromatography. Journal of Chromatographic Science 2021, 59, 412-418.
  5. Wang, W.; Zheng, B.; Wu, J.; Lv, W.; Lin, P.; Gong, X. Determination of the dissociation constants of 16 active ingredients in medicinal herbs using a liquid-liquid equilibrium method. Separations 2021, 8, 1-9.
  6. Liu, Y.; Liu, J.; Wu, K.-X.; Guo, X.-R.; Tang, Z.-H. A rapid method for sensitive profiling of bioactive triterpene and flavonoid from astragalus mongholicus and astragalus membranaceus by ultra-pressure liquid chromatography with tandem mass spectrometry. Journal of Chromatography B-Analytical Technologies in the Biomedical and Life Sciences 2018, 1085, 110-118.
  7. Liu, P.-P.; Shan, G.-S.; Zhang, F.; Shi, J.; Jia, T.-Z. Comparison of 12 constituents of astragali radix directionally processed with organic acid by uplc-ms. Zhongguo Zhong yao za zhi = Zhongguo zhongyao zazhi = China journal of Chinese materia medica 2020, 45, 113-118.
  8. Guo, Z.; Lou, Y.; Kong, M.; Luo, Q.; Liu, Z.; Wu, J. A systematic review of phytochemistry, pharmacology and pharmacokinetics on astragali radix: Implications for astragali radix as a personalized medicine. International Journal of Molecular Sciences 2019, 20, 1-10.
  • This article provides a systematic review of pharmacology and pharmacokinetics and is also an important AR bioactivity reference for this article.
  1. Li, H.; Zhang, Y.; Min, J.; Gao, L.; Zhang, R.; Yang, Y. Astragaloside iv attenuates orbital inflammation in graves' orbitopathy through suppression of autophagy. Inflammation Research 2018, 67, 117-127.
  2. Zhu, Z.; Li, J.; Zhang, X. Astragaloside iv protects against oxidized low-density lipoprotein (ox-ldl)-induced endothelial cell injury by reducing oxidative stress and inflammation. Medical Science Monitor 2019, 25, 2132-2140.
  3. Noest, X.; Pferschy-Wenzig, E.-M.; Nikles, S.; He, X.; Fan, D.; Lu, A.; Yuk, J.; Yu, K.; Isaac, G.; Bauer, R. Identification of constituents affecting the secretion of pro-inflammatory cytokines in lps-induced u937 cells by uhplc-hrms-based metabolic profiling of the traditional chinese medicine formulation huangqi jianzhong tang. Molecules 2019, 24,1-9.
  4. Hao, M.; Liu, Y.; Chen, P.; Jiang, H.; Kuang, H.-Y. Astragaloside iv protects rgc-5 cells against oxidative stress. Neural Regeneration Research 2018, 13, 1081-1086.
  5. Yang, P.; Zhou, Y.; Xia, Q.; Yao, L.; Chang, X. Astragaloside iv regulates the pi3k/akt/ho-1 signaling pathway and inhibits h9c2 cardiomyocyte injury induced by hypoxia-reoxygenation. Biological & Pharmaceutical Bulletin 2019, 42, 721-727.
  6. Murata, I.; Abe, Y.; Yaginuma, Y.; Yodo, K.; Kamakari, Y.; Miyazaki, Y.; Baba, D.; Shinoda, Y.; Iwasaki, T.; Takahashi, K., et al. Astragaloside-iv prevents acute kidney injury and inflammation by normalizing muscular mitochondrial function associated with a nitric oxide protective mechanism in crush syndrome rats. Annals of Intensive Care 2017, 7, 1-9.
  7. Gong, G.; Zheng, Y.; Yang, Y.; Sui, Y.; Wen, Z. Pharmaceutical values of calycosin: One type of flavonoid isolated from astragalus. Evidence-Based Complementary and Alternative Medicine 2021, 2021, 1-9.
  • This article isolates calycosin (flavonoid) from Astragalus membranaceus, an important bioactive study substance discussed in this review.
  1. Yang, B.; Yang, N.; Chen, Y.; Zhu, M.; Lian, Y.; Xiong, Z.; Wang, B.; Feng, L.; Jia, X. An integrated strategy for effective-component discovery of astragali radix in the treatment of lung cancer. Frontiers in Pharmacology 2021, 11, 1-9.
  • This article identifies a comprehensive strategy of active ingredients in Astragalus for the treatment of lung cancer and provides data to support this paper's systematic review of the anti-cancer effects of AR.
  1. Li, W.; Song, K.; Wang, S.; Zhang, C.; Zhuang, M.; Wang, Y.; Liu, T. Anti-tumor potential of astragalus polysaccharides on breast cancer cell line mediated by macrophage activation. Materials Science & Engineering C-Materials for Biological Applications 2019, 98, 685-695.
  2. Xu, C.; Wang, Y.; Feng, J.; Xu, R.; Dou, Y. Extracts from huangqi (radix astragali mongoliciplus) and ezhu (rhizoma curcumae phaeocaulis) inhibit lewis lung carcinoma cell growth in a xenograft mouse model by impairing mitogen-activated protein kinase signaling, vascular endothelial growth factor production, and angiogenesis. Journal of Traditional Chinese Medicine 2019, 39, 559-565.
  3. Zheng, Q.; Zhu, J.-Z.; Bao, X.-Y.; Zhu, P.-C.; Tong, Q.; Huang, Y.-Y.; Zhang, Q.-H.; Zhang, K.-J.; Zheng, G.-Q.; Wang, Y. A preclinical systematic review and meta-analysis of astragaloside iv for myocardial ischemia/reperfusion injury. Frontiers in Physiology 2018, 9, 1-9.
  4. Liu, Z.-h.; Liu, H.-b.; Wang, J. Astragaloside iv protects against the pathological cardiac hypertrophy in mice. Biomedicine & Pharmacotherapy 2018, 97, 1468-1478.
  5. Leng, B.; Tang, F.; Lu, M.; Zhang, Z.; Wang, H.; Zhang, Y. Astragaloside iv improves vascular endothelial dysfunction by inhibiting the tlr4/nf-kappa b signaling pathway. Life Sciences 2018, 209, 111-121.
  6. Lin, X.-P.; Cui, H.-J.; Yang, A.L.; Luo, J.-K.; Tang, T. Astragaloside iv improves vasodilatation function by regulating the pi3k/akt/enos signaling pathway in rat aorta endothelial cells. Journal of Vascular Research 2018, 55, 169-176.
  7. Huang, Y.; Kwan, K.K.L.; Leung, K.W.; Wang, H.; Kong, X.P.; Dong, T.T.X.; Tsim, K.W.K. The extracts and major compounds derived from astragali radix alter mitochondrial bioenergetics in cultured cardiomyocytes: Comparison of various polar solvents and compounds. International Journal of Molecular Sciences 2018, 19.
  8. Qiu, L.-H.; Zhang, B.-Q.; Lian, M.-J.; Xie, X.-J.; Chen, P. Vascular protective effects of astragalus membranaceus and its main constituents in rats with chronic hyperhomocysteinemia. Experimental and Therapeutic Medicine 2017, 14, 2401-2407.
  9. Roden, M.; Shulman, G.I. The integrative biology of type 2 diabetes. Nature 2019, 576, 51-60.
  • This article provides a systematic account of the pathogenesis of type 2 diabetes and provides the basic theoretical support for the discussion in this review.
  1. Cho, J.; D'Antuono, M.; Glicksman, M.; Wang, J.; Jonklaas, J. A review of clinical trials: Mesenchymal stem cell transplant therapy in type 1 and type 2 diabetes mellitus. American Journal of Stem Cells 2018, 7, 82-93.
  2. Hull, C.M.; Peakman, M.; Tree, T.I.M. Regulatory t cell dysfunction in type 1 diabetes: What's broken and how can we fix it? Diabetologia 2017, 60, 1839-1850.
  3. Zhang, R.; Qin, X.; Zhang, T.; Li, Q.; Zhang, J.; Zhao, J. Astragalus polysaccharide improves insulin sensitivity via ampk activation in 3t3-l1 adipocytes. Molecules 2018, 23, 1-9.
  4. Chen, X.; Wang, H.; Jiang, M.; Zhao, J.; Fan, C.; Wang, Y.; Peng, W. Huangqi (astragalus) decoction ameliorates diabetic nephropathy via irs1-pi3k-glut signaling pathway. American Journal of Translational Research 2018, 10, 2491-2501.
  5. Li, M.; Wang, W.; Xue, J.; Gu, Y.; Lin, S. Meta-analysis of the clinical value of astragalus membranaceus in diabetic nephropathy. Journal of Ethnopharmacology 2011, 133, 412-419.
  6. Koh, A.; Backhed, F. From association to causality: The role of the gut microbiota and its functional products on host metabolism. Molecular Cell 2020, 78, 584-596.
  7. Wang, X.-M.; Li, X.-B.; Peng, Y. Impact of qi-invigorating traditional chinese medicines on intestinal flora: A basis for rational choice of prebiotics. Chinese Journal of Natural Medicines 2017, 15, 241-254.
  • This article describes the theory of "qi and blood" and "yin and yang" regulation in Chinese medicine, and provides the basic theoretical support for the discussion of the application of AR in Chinese medicine clinics and its ancient prescriptions in this review.
  1. Yao Y; Zhang H; Li J et al. Efficacy of combining astragalus and turtle shell soup with ear acupressing beans in the treatment of diabetic peripheral neuropathy with Qi and Yin deficiency and the effect on serum myd88/iκb signaling pathway. Chinese Journal of Experimental Formulary. 2021, 27, 98-105. (in Chinese)
  • This article describes the use of ancient Chinese herbal formulas in the treatment of diabetes and provides data to support this review of the use of AR as a major component of ancient Chinese medicine in the treatment of diabetes..
  1. Ul Uza, N.; Dastagir, G. Microscopic and pharmacognostic standardization of astragalus scorpiurus bunge. Microscopy Research and Technique 2022, 85, 324-338.
  • This article provides significant theoretical and data support for the standardisation of AR.

  1. The manuscript lacks in-depth discussion and prospects.

Thank you for your comments. The hypoglycemic effect of AR has been discussed with emphasis based on the theories of "qi and blood" and "yin and yang" in Chinese medicine and modern medical research methods. At the same time, the clinical application of AR as a major component of traditional and important ancient formulas is discussed. Finally, the standardizations of AR is prospected and summarized by us. The details are as follows.

1) The efficacy of AR for cardiovascular disease prevention is summarized and discussed with reference to the existing research articles and reviews. Details are as follows.

Page 19~20, Line 464~490, added

5.5. Cardiovascular disease prevention

An increasing number of experiments have confirmed the effectiveness of AR in inhibiting cardiovascular diseases such as myocardial ischemia-reperfusion injury [72], myocardial hypertrophy [73], vascular endothelial dysfunction [74,75], coronary artery disease [76], atherosclerosis [77], myocardial fibrosis [78], and viral myocarditis [79].

The sarcoplasmic reticulum (SR) plays an essential role in intracellular calcium regulation, and the sarcoplasmic reticulum Ca2+- ATPase (SERCA2a) in the SR is responsible for sarcoplasmic calcium reuptake. Many studies have shown that the expression of SERCA2a is significantly reduced after myocardial injury. AR has been reported to promote SERCA2a expression, enhance calcium reuptake in the SR and inhibit calcium overload, thereby inhibiting oxidative stress-induced apoptosis and cytoskeletal damage in adriamycin-injured rat heart cells [80]. Coronary heart disease (CHD) is another predominant cardiovascular disease and remains a serious public health burden [75]. Endothelium-dependent coronary artery vascular reactivity is a key indicator of vascular function. Endothelial dysfunction is characterised by reduced nitric oxide (NO) bioavailability [81]. In a study, the effects of AR extracts and their main compounds on mitochondrial bioenergetics were evaluated. The results showed that AR aqueous extracts inhibited ROS production in cardiomyocytes under oxidative stress and had the strongest antioxidant activity and protective effect. This protective effect is thought to be mediated by increasing the spare respiratory capacity and mitochondrial ATP production in stressed cells [82]. In addition, metalloproteinases are closely associated with cardiovascular disease caused by hyperhomocysteinemia. The results showed that total AR extract and Astragalus membranaceus total saponin (ASP) increased abnormal aortic NO production, increased superoxide dismutase (SOD) activity, decreased the concentration of metalloproteinases MMP-2 and MMP-9, and significantly reduced abnormal aortic NO production. thereby improving vascular endothelial dysfunction in rats with hyperhomocysteinemia-induced cardiovascular disease [83].

2) The discussion focuses on the hypoglycaemic effects of AR based on the "qi-blood" and "yin-yang" theories of Chinese medicine as well as modern medical research methods. Details are as follows.

Page 21, Line 538~546, added

AR has been shown to be effective in regulating the intestinal flora in the body and in treating type 2 diabetes by improving insulin resistance and lowering blood glucose. According to traditional Chinese medicine, the most important etiological mechanism in T2DM is “Qi and Yin deficiency, heat and toxicity”, which leads to disorders of glucose metabolism, insulin resistance and damage to the intestinal mucosa, all of which can be classified as "heat" and "toxicity" in Chinese medicine [96]. The disorders of glucose metabolism, insulin resistance and intestinal mucous membrane damage can be classified as "heat" and "toxicity" in Chinese medicine. However, the role of the “middle Jiao” in the body includes receiving, decomposing, digesting and absorbing, and the blood is purified so that “evil Qi” can be removed.

3) The clinical application of AR as a major component of traditional and important ancient formulas is discussed. Details are as follows.

Page 21, Line 547~555, added

Wang et al. found that gavage of diabetic rats fed high-fat with the ginseng and astragalus formula (Astragalus, ginseng, smallpox powder, yam, and raw licorice) reduced blood glucose levels, increased the expression of GLP-1 in the serum of the rats, increased the ratio of the Bacteroidetes/Fimicutes, and increased the relative abundance of the Bacteroides, Clostridium butyricum, and Rothia Geory and Brown [96]. Yao et al. used “Astragalus turtle shell soup, ATSS (黄芪鳖甲汤)” to treat T2DM patients and found that the relative abundance of Brucella.spp decreased and that of Bacteroides increased after treatment, indicating that ATSS plays a positive role in regulating the dynamic balance of intestinal flora [97].

4) The standardizations of AR is prospected and summarized. Details are as follows.

Page 21~22, Line 556~593, added

  1. Standardization of AR

There are at least 3 levels of standards regarding the standardization of AR, namely efficacy standards, bioactivity standards, and production standards. AR is complex in origin, substance base, preparation process and quality control indicators, and it is for this reason that there is a lack of established standards for AR, but rather applicable standards must be developed in conjunction with the specific characteristics of specific species [98].

6.1. Efficacy standards

Efficacy criteria are the first standard to be considered for AR, especially in the clinical efficacy. This is determined by its essential properties and cannot be considered in a single, incomplete way by considering its chemical composition content, bioavailability, quality consistency and other modern drug indicators. Of course, efficacy criteria need to be measured through many clinical applications and medical observations, and for this reason, a reasonable evaluation system must be established. Unfortunately, there are few clinical studies on AR, data from clinical samples are lacking, and more studies are only to the extent of cellular or animal experiments.

6.2. Bioactivity standards

The second criterion is the activity criterion, which is considered based on the physical and chemical properties of the AR (such as AR main ingredient content, chemical ingredient content, etc.). It is well known that it is not valid to talk about activity in isolation from dosage. Therefore, the relationship between dose and activity, and between structure and activity, needs to be rigorously studied in order to establish AR activity criteria (such as antioxidant, anti-inflammatory, anti-tumors, anti-viral activity etc.). But what is even more important for AR is batch consistency, as an extract of a natural product, so batch consistency is crucial and arguably a prerequisite for efficacy and safety. Unfortunately, different extraction and isolation methods can have a very significant impact on AR activity, and this poses a major challenge for the development of activity standards.

6.3. Production standards

The production standards mentioned here cover both the cultivation and production of AR. There are no specific standards for AR cultivation and there is a lack of in-depth research. They are generally based on basic standards such as GAP (Good Agricultural Practice) and GMP (Good Manufacturing Practice). But for the future industrialization of AR, more in-depth research is needed for the development of standard production. For example, how does the climate and soil of different production areas affect the accumulation of AR ingredients, and how do the many processes of extraction, alcohol precipitation, extraction and chromatography affect the migration and transformation of target ingredients in the AR production process? The study of the migration and transformation of the target components during the extraction, alcohol deposition, extraction, and chromatography processes.

  1. The format of references is not uniform.

Thank you for your comments. References in the tables and in the full text have been carefully checked by us.

Round 2

Reviewer 3 Report

The manuscript is improved after revision. However, there are several other questions about this manuscript, as follows.

  1. Page 1, line 2~3, there is an additional letter “S” after “AR”.

  1. Regarding the writing of the Latin names of the two proto-plants, it is better to be uniform. For example, Astragalus membranaceus (Fisch.) Bge. var. mongholicus (Bge.) Hsiao was sometimes denoted by A. mongholics.” and sometimes by “A. membranaceus var. mongholicus” in the manuscript.

  1. 457.4 and 617.4 of the 11th and 12th compounds in Table 2 are fragment ions and should not be expressed as [M+H]+.

  1. In Table 3, “+” should be superscripted. The molecular weight of the 13th compound is 300.31. Why is the ion of [M+H]+ 303?

  1. In Tables 2 and 3, some blanks were indicated by “-” and some were not, which is best to keep the same.

  1. Page 15, line 349, what exactly does AS-IV mean? Please add full name.

  1. Page 21, line 591, please change “chemical analysis” to “analytical methods”.

  1. Figure 8 in the manuscript is a direct copy of someone else’s published image. This operation requires authorization and should not be quoted directly.

Author Response

  1. Page 1, line 2~3, there is an additional letter “S” after “AR”.

Thank you for your comment, the error has been corrected due to an oversight on our part. Details are below.

Page 1, Line 33, change “AR s” to “AR”.

  1. Regarding the writing of the Latin names of the two proto-plants, it is better to be uniform. For example, Astragalus membranaceus (Fisch.) Bge. var. mongholicus (Bge.) Hsiao was sometimes denoted by “A. mongholics.” and sometimes by “A. membranaceus var. mongholicus” in the manuscript.

Thank you for your comment. “Astragalus membranaceus (Fisch.) Bge. var. mongholicus (Bge.) Hsiao” has been carefully checked and the expression has been standardized throughout. Details of the revision are given below.

Page 2, Line 43, change to “Figure 1. A. membranaceus and A. var. mongholicu and their commercial application”

Page 2, Line 64, change “A. mongholics” to “A. membranaceus

Page 3, Line 79, in Table 1, change “A. membranaceus var. mongholicus” to “A. membranaceus

Page 3, Line 83~84, change “A. membranaceus var. mongholicus” to “A. membranaceus

Page 5, Line 100, in Table 2, change “A. membranaceus var. mongholicus” to “A. membranaceus

Page 7, Line 120, in Table 3, change “A. membranaceus var. mongholicus” to “A. membranaceus

  1. 457.4 and 617.4 of the 11th and 12th compounds in Table 2 are fragment ions and should not be expressed as [M+H]+.

Thank you for your comment. We deleted the expression of [M+H]+. Details of the modifications can be found at the end.

  1. In Table 3, “+” should be superscripted. The molecular weight of the 13th compound is 300.31. Why is the ion of [M+H]+ 303?

Thank you for your comment. We are so sorry, it is a typo after we checked with the literature carefully. The ion of [M+H]+ for the 13th compound is 301. Details of the modifications can be found at the end.

  1. Page 1, line 2~3, there is an additional letter “S” after “AR”.

Thank you for your comment, the error has been corrected due to an oversight on our part. Details are below.

Page 1, Line 33, change “AR s” to “AR”.

  1. Regarding the writing of the Latin names of the two proto-plants, it is better to be uniform. For example, Astragalus membranaceus (Fisch.) Bge. var. mongholicus (Bge.) Hsiao was sometimes denoted by “A. mongholics.” and sometimes by “A. membranaceus var. mongholicus” in the manuscript.

Thank you for your comment. “Astragalus membranaceus (Fisch.) Bge. var. mongholicus (Bge.) Hsiao” has been carefully checked and the expression has been standardized throughout. Details of the revision are given below.

Page 2, Line 43, change to “Figure 1. A. membranaceus and A. var. mongholicu and their commercial application”

Page 2, Line 64, change “A. mongholics” to “A. membranaceus

Page 3, Line 79, in Table 1, change “A. membranaceus var. mongholicus” to “A. membranaceus

Page 3, Line 83~84, change “A. membranaceus var. mongholicus” to “A. membranaceus

Page 5, Line 100, in Table 2, change “A. membranaceus var. mongholicus” to “A. membranaceus

Page 7, Line 120, in Table 3, change “A. membranaceus var. mongholicus” to “A. membranaceus

  1. 457.4 and 617.4 of the 11th and 12th compounds in Table 2 are fragment ions and should not be expressed as [M+H]+.

Thank you for your comment. We deleted the expression of [M+H]+. Details of the modifications can be found at the end.

  1. In Table 3, “+” should be superscripted. The molecular weight of the 13th compound is 300.31. Why is the ion of [M+H]+ 303?

Thank you for your comment. We are so sorry, it is a typo after we checked with the literature carefully. The ion of [M+H]+ for the 13th compound is 301. Details of the modifications can be found at the end.
